# Estimating dry biomass and plant nitrogen concentration in pre-Alpine grasslands with low-cost UAS-borne multispectral data – a comparison of sensors, algorithms, and predictor sets

Anne Schucknecht[1], Bumsuk Seo[1], Alexander Krämer[2], Sarah Asam[3], Clement Atzberger[4], Ralf Kiese[1]

[1]Institute of Meteorology and Climate Research – Atmospheric Environmental Research (IMK-IFU), Karlsruhe Institute of Technology (KIT), Garmisch-Partenkirchen, 82467, Germany
[2]WWL Umweltplanung und Geoinformatik GbR, Bad Krozingen, 79189, Germany
[3]German Aerospace Center, German Remote Sensing Data Center, 82234 Wessling, Germany
[4]Institute of Geomatics, University of Natural Resources and Life Sciences (BOKU), Vienna, 1190, Austria

*Correspondence to*: Anne Schucknecht (anne.schucknecht@kit.edu)

**Abstract.** Grasslands are an important part of pre-Alpine and Alpine landscapes. Despite the economic value and the significant role of grasslands in carbon and nitrogen (N) cycling, spatially explicit information on grassland biomass and quality is rarely available. Remotely sensed data from unmanned aircraft systems (UAS) and satellites might be an option to overcome this gap. Our study aims to investigate the potential of low-cost UAS-based multispectral sensors for estimating above-ground biomass (dry matter, DM) and plant N concentration. In our analysis, we compared two different sensors (Parrot Sequoia, SEQ; MicaSense RedEdge-M, REM), three statistical models (Linear Model; Random Forests, RF; Gradient Boosting Machines, GBM) and six predictor sets (i.e. different combinations of raw reflectance, vegetation indices, and canopy height). Canopy height information can be derived from UAS sensors, but was not available in our study. Therefore, we tested the added value of this structural information with in-situ measured bulk canopy height data. A combined field sampling and flight campaign was conducted in April 2018 at different grassland sites in Southern Germany to obtain in-situ and the corresponding spectral data. The hyper-parameters of the two machine learning (ML) approaches (RF, GBM) were optimized and all model set-ups were run with a six-fold cross-validation. Linear models were characterized by very low statistical performance measures, thus were not suitable to estimate DM and plant N concentration using UAS data. The non-linear ML algorithms showed an acceptable regression performance for all sensor-predictor set combinations with average (avg) $R^2_{cv}$ of 0.48, $RMSE_{cv, avg}$ of 53.0 g m² and $rRMSE_{cv, avg}$ of 15.9% for DM, and with $R^2_{cv, avg}$ of 0.40, $RMSE_{cv, avg}$ of 0.48 wt.% and $rRMSE_{cv, avg}$ of 15.2% for plant N concentration estimation. The optimal combination of sensors, ML algorithms and predictor sets notably improved the model performance. The best model performance for the estimation of DM ($R^2_{cv}$ = 0.67, $RMSE_{cv}$ = 41.9 g m², $rRMSE_{cv}$ = 12.6%) was achieved with a RF model that utilizes all possible predictors and REM sensor data. The best model for plant N concentration was a combination of a RF model with all predictors and SEQ sensor data ($R^2_{cv}$ = 0.47, $RMSE_{cv}$ = 0.45 wt.%, $rRMSE_{cv}$ = 14.2%). DM models with the spectral input of REM performed significantly better than those with SEQ data, while for N concentration models it was the other way round. The choice of predictors was most influential on model performance, while the effect of the chosen ML algorithm was generally lower. The addition of canopy

height to the spectral data in the predictor set significantly improved the DM models. In our study, calibrating ML algorithm improved the model performance substantially, which shows the importance of this step.

## 1 Introduction

Grasslands are import ecosystems covering about 40% of the global land area (excluding Antarctica and Greenland) (White et al., 2000). In pre-Alpine (i.e. the hilly Alpine foreland) and Alpine landscapes (i.e. the core Alps), grasslands are a dominant element. (Pre-)Alpine grassland ecosystems provide a variety of goods and services (Egarter Vigl et al., 2016) such as food and forage for livestock production, leading to a high economic value (Egarter Vigl et al., 2018; Gibson, 2009; White et al., 2000). At the same time, grassland plants and soils play a significant role in carbon (C) and nitrogen (N) cycling (Gibson, 2009; Wiesmeier et al., 2013), and are improving water purification and soil stability (Lamarque et al., 2011). Furthermore, mountain grasslands are among the most species-rich ecosystems in Europe and high in endemism (Ewald et al., 2018; Väre et al., 2003; Veen et al., 2009; White et al., 2000). With the agricultural intensification in the lowlands, mountain grasslands act increasingly as sanctuary for species that were common throughout Europe (European Environmental Agency, 2010). Therefore, grasslands in mountain areas have important environmental, biological as well as aesthetic functions (Fontana et al., 2014).

Besides changing climatic conditions, human intervention proofed to be an equally important driver to changing ecosystem functioning in managed (pre-)Alpine grasslands (Rossi et al., 2020; Schirpke et al., 2017; Spiegelberger et al., 2006; Walter et al., 2012). The knowledge about grassland yields (biomass) and fodder quality is critical for the management of grasslands and livestock, e.g. with regard to harvest time and frequency, stocking rates, or timing and amount of fertilizer application (Capolupo et al., 2015; Primi et al., 2016). Grassland quality with respect to the nutritive value of forage is assessed by key chemical parameters including crude protein or N, fibre, organic matter digestibility (OMD), and metabolisable energy (ME) (Pullanagari et al., 2016, 2013).

On the field scale, information needs of farmers are closely related to different national implementations of the European Nitrates Directive (Council Directive 91/676/EEC of 12 December 1991), influencing management practices and economic revenues. On a regional scale, ecosystem characteristics such as the N balance and associated losses of greenhouse gases and N leaching needs to be assessed by authorities.

N uptake by plants is the highest N flux in pre-Alpine grasslands (Schlingmann et al., 2020; Zistl-Schlingmann et al., 2020). Thus, N uptake in relation to fertilization rates represents an important measure for optimizing grassland management on farm and regional scale, as decision-making is getting more and more complex due to legislation and climate change (e.g. drought effects). Hence, a thorough mapping, monitoring and assessment of grassland traits such as above-ground biomass (dry matter, DM) and chemical composition parameters (e.g. plant N concentration) is required to ensure the preservation of grassland ecosystems and their sustainable use. However, spatially explicit and accurate information on grassland biomass and quality

at field and regional scale is lacking. Robust and reliable methods and applications for grassland monitoring are needed, which

ideally scale well and are cost-effective.

Considering the diversity and the large area covered by grasslands, traditional techniques based on field sampling or proximal sensing (e.g. field spectrometers) reach their limits when aiming for a regional assessment of grassland traits (Wachendorf et al., 2017). Here, remotely sensed data from satellites are increasingly established as promising data sources for a continuous and comprehensive mapping of vegetation parameters. Green vegetation can be monitored continuously using its spectral

reflectance properties acquired by optical sensors (Atzberger, 2013; Baret and Buis, 2008). The utilization of satellite information is of high value in particular when large and/or remote areas need to be studied. Also the fast data collection and processing, the relatively low costs of many remote sensing data products (Wachendorf et al., 2017) as well as time series of well calibrated satellite sensors are advantageous.

However, while emerging services such as the Copernicus Land Monitoring Services provide land cover information at an

unprecedented spatial and temporal resolution, these products still do not provide the necessary spatially detailed information in specific areas such as mountain regions. Mountains are often characterized by small and heterogeneous grassland patches, a high overall cloud occurrence, and frequent cloud formation at specific locations. Furthermore, steep terrain leads to shadows often affecting permanently the same areas given the constant acquisition time of most satellites. Even outside permanently shadowed areas, bidirectional reflectance distribution function (BRDF) effects result from the highly variable sun-sensor-

terrain geometries (Richter, 1998). Together, these factors limit the reliability of space-borne observations in mountainous areas. Airborne remote sensing data has occasionally been used in the past to match the required spatial scale and to explore the increased radiometric resolution of hyperspectral sensors (Atzberger et al., 2015; Burai et al., 2015; Darvishzadeh et al., 2011). But airborne data are still affected by the above mentioned weather and topography related challenges. Furthermore, they are associated with higher costs for the users if there is no data available for the study region from other flight campaigns.

Remotely sensed data from unmanned aircraft system (UAS) are a promising possibility to overcome satellite and airborne-specific issues due to their high flexibility in flight planning, the very high spatial resolution (lower cm range, depending on flight height) and the availability of some low-cost multispectral systems. Vegetation traits can be mapped under challenging conditions at the field scale applying UAS (Maes and Steppe, 2019). BRDF information can be derived from UAS sensors - similar to traditional airborne campaigns - as data are usually flown with high overlap, providing additional information

(Koukal and Atzberger, 2012). However, besides their advantages, data acquisition with UAS has also some limitations. Most UAS cannot be operated under moist and windy conditions and legal restrictions of the country and study regions need to be considered. Changing illumination (e.g. through clouds and variations in solar angle) affects the quality of imagery making a sound radiometric calibration an essential processing step. Accordingly, the standardization and comparability of sensors and workflows is an issue, especially when accounting for the quality of low-cost sensors (Aasen et al., 2018; Assmann et al.,

2018; Olsson et al., 2021; Poncet et al., 2019; Salamí et al., 2014).

Previous studies using UAS data have looked into the mapping of biophysical parameters such as Leaf Area Index (LAI) (Verger et al., 2014; Yao et al., 2017), chlorophyll (Jay et al., 2017), biomass (Näsi et al., 2018; Viljanen et al., 2018), plant

density (Jin et al., 2017), canopy height (Song and Wang, 2019; Ziliani et al., 2018) as well as combinations of these parameters (Jay et al., 2019). However, most UAS studies investigate the mapping of plant traits in monocultural crop stands, while multispecies systems such as natural or cultivated permanent grassland ecosystems like in pre-Alpine regions have been studied less often. Notable exceptions are Bareth and Schellberg (2018), Grüner et al. (2019), Lussem et al. (2019), Wang et al. (2017), and Zhang et al. (2018). Even fewer studies investigate the potential of UAS-borne sensor data for the estimation of grassland quality. Capolupo et al. (2015) estimated various biochemical plant traits (crude protein, crude ash, crude fiber, sodium and potassium concentration, metabolic energy) from UAS-acquired hyperspectral images (400–950 nm) of experimental grassland plots in Germany. The authors compared the use of linear regression with narrowband vegetation indices (VI) and partial least squares regressions (PLSR), concluding that PLSR yielded better results for biochemical parameters ($R^2$ ranging from 0.21 for sodium until 0.80 for metabolic energy). Wijesingha et al. (2020) investigated crude protein and acid detergent fibre of eight grassland sites in Hesse (Germany) using a hyperspectral sensor (450–998 nm). Five predictive regression algorithms were tested, of which the support vector regression achieved the best result for crude protein estimation (normalized RMSE = 10.6%), and a cubist regression model proved best for acid detergent fibre estimation (normalized RMSE = 13.4%). Although these studies achieved promising results for forage quality estimation, they rely on hyperspectral data.

There are far fewer studies available utilizing cheaper UAS-borne multispectral data to estimate grassland quality parameters. Caturegli et al. (2016) utilized the NDVI calculated from multispectral sensor (Tetracam ADCMicro) data in a linear regression to estimate the N status of three turfgrass species. Depending on the species, $R^2$ varied between 0.66 and 0.86. Hence, the potential of low-cost multispectral UAS-borne data for field-scale mapping and assessment of multispecies grasslands is not yet fully tested and exploited.

Thus, the objective of this study is to evaluate the potential of low-cost UAS data for estimating DM and plant community N concentration of managed pre-Alpine grasslands. The multispectral Parrot Sequoia sensor (SEQ) has been applied in several vegetation mapping/monitoring studies in the agricultural context (e.g., Grüner et al., 2020; Guan et al., 2019; Handique et al., 2017; Matsumura, 2020; Moncayo-Cevallos et al., 2018; Stroppiana et al., 2018). However, some associated quality issues have been reported (Olsson et al., 2021; Poncet et al., 2019). Therefore, we want to compare the performance of the SEQ sensor with another low-cost multispectral sensor, namely the MicaSense RedEdge-M (REM). We used statistical learning algorithms to build regression models and estimated DM and N over the whole UAS scenes. We utilized the multispectral data of the two UAS sensors together with in-situ data of DM, N concentration and bulk canopy height (CH) from a test campaign in April 2018 on sites in Southern Germany. Additionally to the multi-spectral data, we evaluated the importance of canopy height as a predictor, primarily to see if it could improve the predictive performance of the models for our study region. In our study, we addressed the following research questions:

 (i) Is the spectral information of the UAS sensors sufficient to estimate and map the spatial pattern of DM and N concentration on managed pre-Alpine grasslands?

 (ii) How important is a calibration of hyper-parameters of the tested machine learning algorithms for the model performance?

(iii) What are the effects of different sensors, statistical modelling approaches and predictor sets on the predictive capabilities of the models?

## 2 Material and methods

### 2.1 Study area, sampling design, and measurements of grassland traits

The study area is located in Southern Germany (Fig. 1), within the German Terrestrial Environmental Observatories (TERENO) Pre-Alpine Observatory (Kiese et al., 2018; Zacharias et al., 2011). The region is characterized by a warm temperate climate i.e. Cfb climate zone according to the Köppen-Geiger climate classification (Rubel et al., 2017). For the period 1981-2010 the mean annual air temperature at the study sites was between 8.0°C and 8.6°C (DWD Climate Data Center, 2019b), and mean annual precipitation between 1008 mm and 1419 mm (DWD Climate Data Center, 2019a). Field data was acquired at ten plots on managed grasslands (Table 1). The plots are situated on the three sites "Fendt" (FE, 600 m a.s.l.), "Rottenbuch" (RB, 700 m a.s.l.), and "Eschenlohe" (EL, 630 m a.s.l.). Care was taken to include different grassland types and management practices in order to render robust and transferable models also for our single campaign. The plots represent a variety of management intensities ranging from very extensively managed grasslands with no fertilizer application and just one cut per year to very intensively managed grasslands with five cuts and five slurry applications per year. A species inventory in June 2020 characterized nine out of ten plots as Arrhenatheretum elatioris while one was classified as Caricion davallianae grasslands (Table 1). Fig. 2 provides an overview of the workflow of this study. Details on the different working steps are presented in the following paragraphs and chapters.

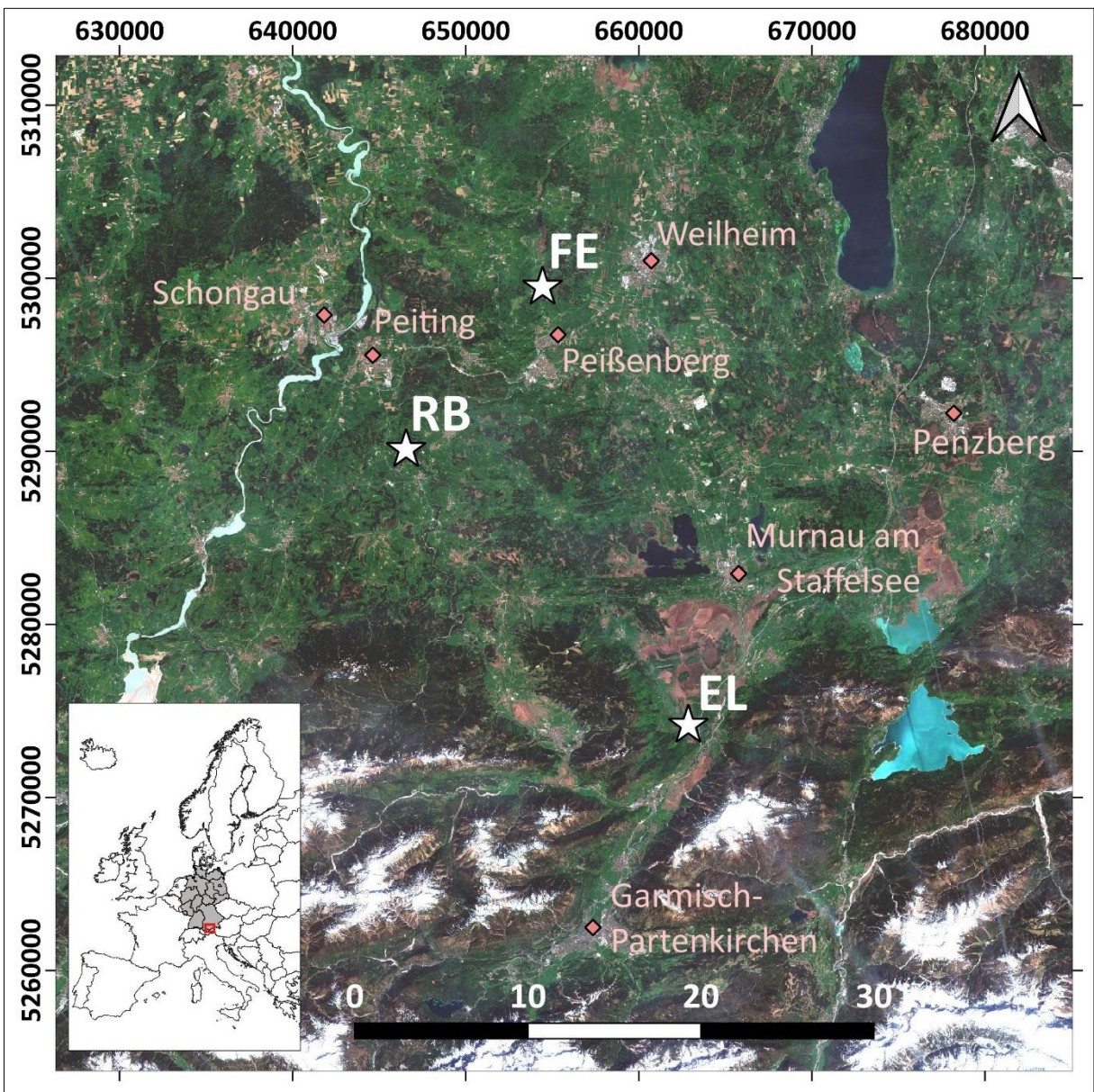

**Figure 1.** Location of the three study sites (white stars) in the study area in the South of Germany. EL = Eschenlohe, FE = Fendt, RB = Rottenbuch. Major towns are indicated for reference (pink diamonds). Background: true colour composite of Sentinel 2B images from 27/04/2018 (contains modified Copernicus Sentinel data [2018], processed by ESA). Used coordinate reference system: EPSG: 25832.

The field campaign with UAS flights and vegetation sampling took place from 24-25 April 2018. The phenological stage of

155 the plots ranged from the principal growth stage 1 (leaf development) to 4 (development of harvestable vegetative plant parts) (Table 1). After the UAS flights, at each site (FE, RB, EL) up to four 30 m x 30 m plots (FE1, FE2, F3, FE4, RB1, RB2, RB3, EL1, EL2, EL3) were sampled at nine to twelve georeferenced subplots of 0.25 m x 0.25 m. Bulk canopy height (CH, in cm)

was measured with a rising plate meter. The vegetation within the subplot was clipped down to stubble height (3 cm). In the lab, the vegetation samples were sorted into the plant functional types non-green vegetation, legumes, non-leguminous forbs and graminoids. After the samples were dried in an oven at 65°C until constant weight was achieved, the dry weight was determined and the dry biomass per area calculated (dry matter, DM, in g m$^{-2}$). For the determination of mean plant community nitrogen concentration (plant N concentration, mass-based, in wt.%), the dried vegetation samples were milled and analysed with an elemental analyser (varioMax CUBE, Elementar Analysesysteme GmbH, Germany). The reader is referred to the corresponding data paper (Schucknecht et al., 2020a) for more detailed information on the sampling, sample processing and analysis.

**Table 1.** Site and plot characteristics partly taken from Schucknecht et al. (2020a). Mean annual climate parameters (MAP = Mean annual precipitation height; MAT = Mean annual temperature) were derived from the DWD Climate Data Center (DWD Climate Data Center, 2019a, b) and correspond to the period 1981-2010. Grassland type and species richness (SR; i.e. number of vascular plant species) were obtained by a species inventory in 2020 (Schuchardt and Jentsch, 2020). The phenological stage was determined by inspecting the photos of the plots with respect to the dominant species (species abbreviations: LM = *Lolium multiflorum*, TR = *Trifolium repens*, LP = *Lolium perenne*, PP = *Poa pratensis*, KP = *Koeleria pyramidata*, FP = *Festuca pratensis*). Provided is the number of the principle growth stage (1 = leaf development (main shoot), 2 = formation of side shoots/ tillering, 3 = stem elongation or rosette growth/ shoot development (main shoot), 4 = development of harvestable vegetative plant parts or vegetatively propagated organs/ booting (main shoot)) according to the BBCH classification (Meier, 2018)

| Site/Plot | Elevation [m a.s.l.] | MAP [mm] | MAT [°C] | Management | Grassland type | SR | Phenological stage |
|---|---|---|---|---|---|---|---|
| Fendt (FE) | 600 | 1008 | 8.6 | | | | |
| FE1 | | | | 5 cuts, no pasture, 4x slurry | Arrhenatheretum elatioris | 20 | LM: 3 |
| FE2 | | | | 4 cuts, no pasture, 3x slurry | Arrhenatheretum elatioris | 15 | LM: 3 |
| FE3 | | | | 5 cuts, no pasture, 4x slurry | Arrhenatheretum elatioris | 17 | LM: 3 |
| FE4 | | | | 5 cuts, no pasture, 4x slurry | Arrhenatheretum elatioris | 19 | TR: 2 |
| Rottenbuch (RB) | 750 | 1159 | 8.0 | | | | |
| RB1 | | | | 3-4 cuts, pasture, 4-5x slurry | Arrhenatheretum elatioris | 30 | LP: 3 |
| RB2 | | | | 5 cuts, no pasture, 5x slurry | Arrhenatheretum elatioris | 25 | PP: 3 |
| RB3 | | | | 1 cut, no pasture, no slurry | Caricion davallianae | 44 | KP: 1 |
| Eschenlohe (EL) | 630 | 1419 | 8.0 | | | | |
| EL1 | | | | 1 cut, pasture, 2x slurry | Arrhenatheretum elatioris | 17 | LP: 3 |
| EL2 | | | | 4 cuts, no pasture, 4x slurry | Arrhenatheretum elatioris | 23 | LP: 3 |
| EL3 | | | | 3 cuts, no pasture, 2x slurry | Arrhenatheretum elatioris | 27 | FP: 3-4 |

From the collected in-situ data we used the information from the single subplots to develop the models (see chapter 2.3). Canopy height (CH) was used as a predictor variable, and DM and plant N concentration as response variables (Fig. 2).

**2.2 Acquisition and (pre-)processing of UAS-borne data**

**2.2.1 UAS flights**

Two different multispectral sensors were tested for this experiment: the four-band Parrot Sequoia (SEQ; Parrot Drones SAS, Paris, France) and the five-band MicaSense RedEdge-M (REM; MicaSense Inc., Seattle, USA) (Table 2). For measuring the incoming solar radiation, both sensors were accompanied by irradiance sensors ("sunshine sensors") that were attached at the top of the drones. This information was used for image-calibration during data processing. Before each flight, data from sensor-

185 specific calibration targets were taken for radiometric calibration of the multispectral images during the processing.

The UAS flights over the FE and RB sites took place on 24/04/2018 between 09:50 and 16:30 and the ones over the EL site (EL-North and EL-South) on 25/04/2018 between 09:00 and 10:50. The SEQ was operated on a fixed-wing UAS (eBee, senseFly, Cheseaux-sur-Lausanne, Switzerland) with automated flight control. The flight height was set to 80 m leading to a ground sample distance of 8.7 – 12.9 cm (depending on the terrain relief). The eBee was flown with a regular grid flight pattern

with an image overlap of 75%.

The REM was operated on a multicopter UAS (DJI Matrice 200, SZ DJI Technology Co., Ltd., Shenzhen, China) by an external company (Globe Flight GmbH, Germany). Due to logistical reasons only the FE and RB sites could be covered. The multicopter was flown manually on a flight height of about 70 m following a regular grid with an overlap of the single images of > 80%. The ground sample distance of the different REM flights was between 7.7 – 8.8 cm.

For all flights with the different sensors, up to 10 Ground Control Points (GCPs) were distributed in the overflight area of the UAS for georeferencing. The exact coordinates of the GCPs` centres were obtained with a Global Navigation Satellite System (GNSS) receiver (Viva GNSS GS 10, Leica Geosystems AG, Switzerland) run in static mode for 10 minutes which resulted in an accuracy of 0.3 cm in horizontal direction and 0.5 cm in vertical direction in post-processing mode (Datasheet of Leica Viva GNSS GS10 receiver, 2020).

**Table 2. Details about the two multispectral sensors used in this study.**

| Parameter | Parrot Sequoia (SEQ) | MicaSense RedEdge-M (REM) |
|---|---|---|
| **Spectral resolution** [nm] | | |
| central wavelength \| band width | | |
| Blue | n.a. | 475 \| 20 |
| Green | 550 \| 40 | 560 \| 20 |
| Red | 660 \| 40 | 668 \| 10 |
| Red edge | 735 \| 10 | 717 \| 10 |
| NIR | 790 \| 40 | 840 \| 40 |
| **Detector size**, x, y [mm] | 4.8 x 3.6 | 4.8 x 3.6 |
| **Number of recorded pixel**, x, y | 1280 x 960 | 1280 x 960 |
| **Lens** | | |
| Focal length of lens [mm] | 4 | 5.5 |
| Aperture (f-number) | 2.2 | 2.8 |

### 2.2.2 Processing of UAS images

The processing of the UAS images was done with the Pix4dMapper Pro software (Pix4D S.A., Prilly, Switzerland) and consisted of three steps. The photogrammetric processing was based on a structure from motion (SfM) approach. First, keypoints of the images were extracted and matched and the internal (e.g. focal length) and external (e.g. orientation) parameters of the camera were calibrated. Georeferencing was done with the integration of the measured GCPs and their identification on several input pictures. The root mean square error (RMSE) of the georeferencing varied between 1.9 cm and 4.7 cm according to the Pix4d processing reports. As a result of the first step, georeferenced automated tie points were created. In the second step, the point cloud densification was done corresponding to the Pix4D-standard-template for agricultural applications. The final step included the mosaicking of the adjusted and calibrated single images to the orthomosaics of each single band. The final spatial resolution of the multispectral images was 9.6 cm for FE, 10.2 cm for RB-North, 10.0 cm for RB-South, 8.7 cm for EL-North, and 12.9 cm for EL-South for the SEQ data, and 7.7 cm for FE, 8.8 cm for RB-North, and 8.2 cm for RB-South for REM data. The radiometric correction of the input-images was done using the data of the irradiance sensor and the reflectance panels.

Additional flights of the fixed-wing UAS equipped with an RGB camera (Sony Cyber-shot WX 220, Sony Corp., Minato, Japan) were performed on all sites to retrieve higher resolution orthophotos (spatial resolution: 0.030 m to 0.043 m) for the different sites of the study area. The georeferenced high resolution orthophotos were used to manually extract the coordinates of the centre points of the subplots (Schucknecht et al., 2020a). Afterwards, the reflectance values of the georeferenced multispectral images from SEQ and REM were extracted and averaged for each subplot using a 3 by 3 pixel window around the centre point (Fig. 2, grey box). The 3 by 3 pixel window approximately corresponds to the size of the subplot. Due to the

high horizontal accuracy of the GNSS measurements (0.3 cm) and the low RMSE of georeferncing (max. 4.7 cm) we expect just minor location errors.

Note that we could just use spectral information from the obtained UAS images as predictors in the model development. Theoretically, it is also possible to derive canopy height information from high-resolution UAS-derived RGB data by creating a digital surface model and subtracting the digital terrain model (DTM) from it as e.g. shown by Grüner et al. (2019) and Wijesingha et al. (2019). Poley and McDermid (2020) emphasized the importance of a high-quality DTM for deriving reliable vegetation structure estimates from UAS imagery. Unfortunately, we did not have such a high-quality DTM for our study sites and hence could not derive UAS-based canopy height information. Therefore, we used the in-situ bulk CH as a substitute to build models with CH as a predictor variable.

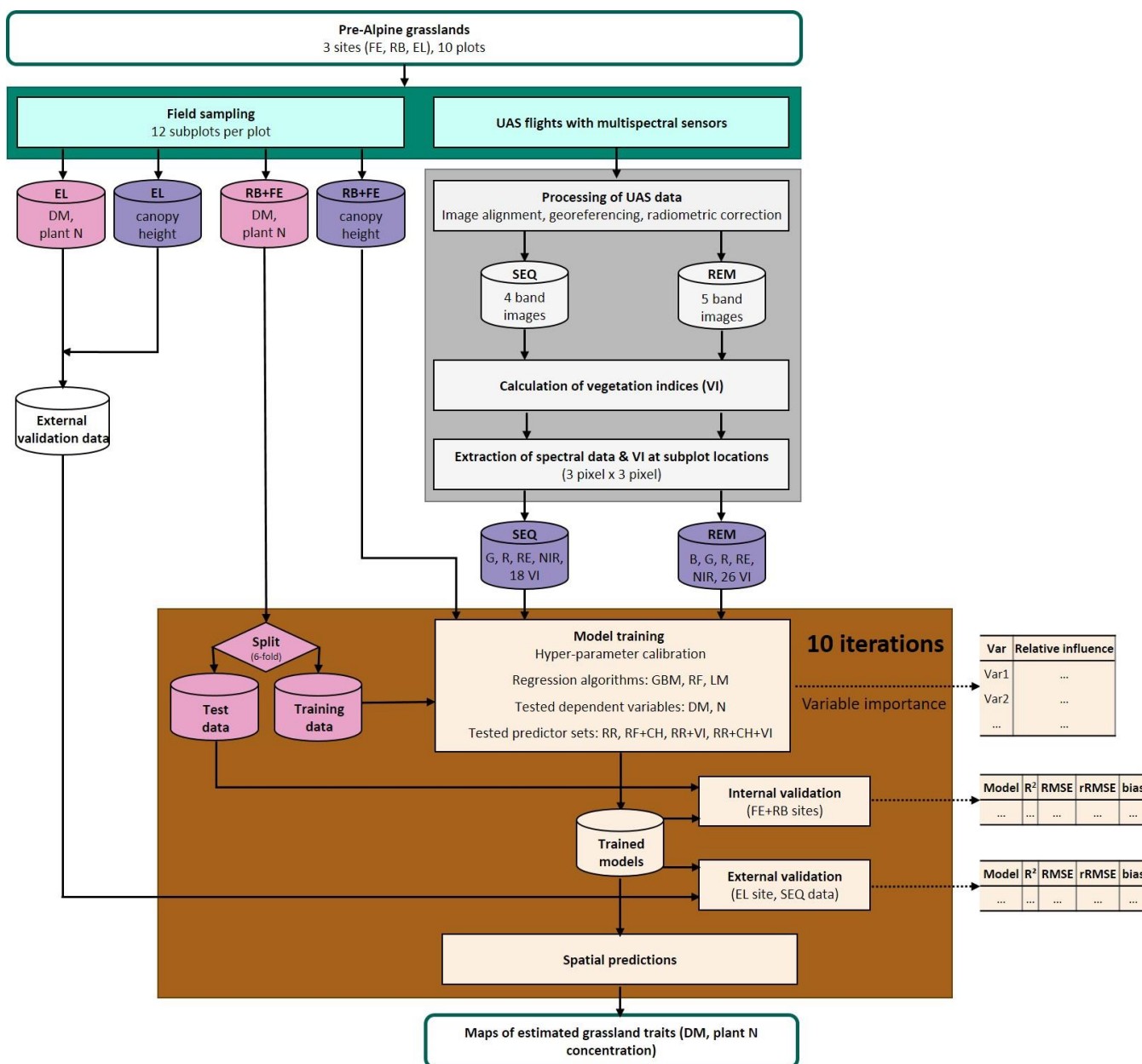

**Figure 2.** Workflow of data acquisition in the field (blue-green), spectral data processing (grey), and model building, validation and application (brown). Response variables of the model are shown in pink and predictor variables in violet. Explanation of abbreviations: sampling sites (FE = Fendt, RB = Rottenbuch, EL = Eschenlohe); predictor variables (G = green band, R = red band, RE = red edge band, NIR = near infrared band, VI = vegetation indices, CH = canopy height); dependent variables (DM = dry matter, N = nitrogen concentration); 235 regression algorithms (GBM = Gradient Boosting Machines, RF = Random Forests, LM = linear models)

### 2.2.3 Vegetation indices

A set of different vegetation indices (VI) was calculated from the spectral bands (Supplementary Table ST1). The various ratio (number of indices used n = 6), orthogonal (n = 1), hybrid (n = 5), red edge (n = 4), and modified chlorophyll indices (n = 4) were selected from the overview presented in Asam (2014). In addition, hyperspectral indices dedicated to chlorophyll (n = 6) were selected from the summary of Ollinger (2011) and adapted to the multispectral data. In total, 26 VI were calculated for REM data and 18 for SEQ data (due to the missing blue band).

### 2.3 Model specifications for DM and plant N concentration estimation

### 2.3.1 Model selection

Regression models were built to estimate DM and plant N concentration based on multispectral UAS data and in-situ bulk canopy height information (Fig. 2, brown box corresponding to model building, validation and application). Combinations of several regression algorithms and predictor sets (PS) were compared to see how different modelling schemes affect the model performance. Two machine learning (ML) algorithms, namely Gradient Boosting Machines (GBM; Friedman, 2002, 2001) and Random Forests (RF; Breiman, 2001), were used in this study. They have been confirmed to be comparable to the other state-of-the-art (classic) machine learning methods for remote sensing applications (Caruana and Niculescu-Mizil, 2006; Fernández-Delgado et al., 2019, 2014; Orzechowski et al., 2018). The two selected algorithms are ensemble-based and have a relatively small number of hyper-parameters (Bernard et al., 2009; Friedman, 2001; Probst et al., 2019). These ensemble-based ML algorithms are known to be able to deal with a large number of highly correlated features (e.g. spectral data and derived vegetation indices) and non-linear relationships without excessive data pre-processing (Hengl et al., 2018). In addition to them, linear regression model (LM) was built to serve as a baseline statistical learning model in the model performance comparison. GBM (Friedman, 2002, 2001) is an ensemble of models based on the idea that weak learners can form a strong learner. The algorithm is adding weak models using a gradient descent process. Gradient boosting can take various forms i.e. different loss functions and optimization schemes. In this study, we took the standard implementation from Friedman (2001, 2002) following Greenwell et al. (2020). GBM has a few tunable parameters, with the major parameters including number of trees ($N_{tree}$), learning rate, and interaction depth (see Table 3), which are supposed to be calibrated using domain data, to avoid under- and over-fitting (Greenwell et al., 2020).

RF is a decision-tree-based ensemble algorithm that uses bootstrap aggregation (i.e., bagging) and the random subspace method (Breiman, 2001). For each decision tree a new bootstrap sample of the training data is created and the tree is fitted to the data. RF has three hyperparameters, namely the number of trees ($N_{tree}$), the number of randomly selected predictors in each split of the decision tree ($m_{try}$) and the minimum number of samples in terminal nodes (node size). It is suggested that for a good model performance the number of trees need to be large enough, but should not yield to overfitting (Strobl et al., 2009). The parameter $m_{try}$ should be carefully calibrated, in particular when predictors are strongly correlated (e.g., Bernard et al., 2009; Kuhn and

Johnson, 2013; Probst et al., 2019; Strobl et al., 2009). The node size determines how many samples a tree needs to grow without being pruned.

### 2.3.2 Hyper-parameter calibration

We parametrized the machine learning algorithms using the nested cross-validation scheme (Arlot and Celisse, 2010; Vabalas et al., 2019; Varma and Simon, 2006) (Table 3). In the nested design, the optimizer in the calibration routine does not see the information included in the hold-out fold. The calibration is done for each of the 10 iterations for randomly split 6 cross-validation folds. For each training fold, parameter searching was done in an internal 5-fold cross-validation using the root mean square error (RMSE) as a penalty function.

To minimize the computing time, we used an efficient parameter space searching algorithm, named (Sequential) Model Based Optimization (MBO) (Bischl et al., 2014; Martinez-Cantin et al., 2007; Shahriari et al., 2016). In this algorithm, an optimizer traverses the parameter space guided by a naive Bayesian parameter proposal function, which identifies a candidate region that is likely to include the optimal parameter combinations. In its iterative process, a new parameter proposal is made based on an acquisition function, or `*infill'*, which is supposed to offer the best improvement in the next step. We used the `*confidence*

*bound'* as infill for GBM and RF. This infill proposes a parameter combination to minimize uncertainty around the good parameter estimates (Bischl et al., 2014). It tries to evaluate the parameter region with large uncertainty with low errors (i.e. good performance), thus expects to reach a large improvement if searched in the next iteration. In this study, parameter values are proposed and evaluated in 500 iterations sequentially and the final values are selected by the lowest error. The impact of calibration is quantified by the difference between the initial error (i.e., based on the random combination of the parameters

sampled from the prescribed ranges) and the best error, which is defined by the lowest error achieved (Malkomes et al., 2016; Swersky et al., 2013). Note that the calibration was done for the 10 iterations individually, in each iteration the nested six-folds share the calibrated parameters. Calibrated values and their summaries are presented in Supplementary Table ST2 and Supplementary Fig. SF2 and SF3.

**Table 3.** Range of the hyper-parameters used in the calibration for Gradient Boosting Machines (GBM) and Random Forests (RF). The calibration routine searches the optimal parameter values within the prescribed ranges. Typical default values for GBM from Greenwell et al. (2020) and RF from Probst et al. (2019). The final calibrated hyper-parameters are presented in Supplementary Table ST2.

| Algorithm | Parameter | Description | Range | Typical default values |
|---|---|---|---|---|
| GBM | Shrinkage | Learning rate (high values may introduce sub-optimal performance, low values slow learning) | [0, 1] | 0.01 to 0.1 |
| | Interaction depth | Maximum level of variable interactions | [1,…,6] | 3 |
| | $N_{tree}$ | Number of trees | [2E3,…, 5E4] | 1000 |
| RF | $m_{try}$ | Number of randomly selected variables on each split | [1,…, $N_{predictors}$/2] | $N_{predictors}$/3 |
| | Node size | Minimum number of samples in terminal nodes | [1,…,5] | 5 |
| | $N_{tree}$ | Number of trees | [5E2,…, 1E4] | 1000 |

### 2.3.3 Predictor set definition

Six different sets of predictor combinations were used in the models. The number of predictors differs for models using SEQ and REM data and is provided in parenthesis below:

- PS1: Raw reflectance bands: using only raw reflectance data from SEQ (n = 4) and REM (n = 5), baseline scenario

- PS2: Vegetation indices (VI): using just VI, but not raw reflectance bands ($n_{SEQ}$ = 18, $n_{REM}$ = 26)

- PS3: Raw reflectance bands and vegetation indices (VI) ($n_{SEQ}$ = 22, $n_{REM}$ = 31)

- PS4: Bulk canopy height (CH, from field measurements): testing the sole use of CH as a reference for structural information (n = 1)

- PS5: Raw reflectance bands and bulk canopy height (CH, from field measurements): using spectral and structural information (CH) ($n_{SEQ}$ = 5, $n_{REM}$ = 6)

- PS6: Raw reflectance bands, CH, and VI: all available spectral and structural input data ($n_{SEQ}$ = 23, $n_{REM}$ = 32)

Bulk CH was selected as a predictor, because we wanted to test the effect of adding structural information, i.e. can the addition of UAS-derived structural information to the spectral information improve the estimation of DM and N concentration in pre-Alpine grasslands? Due to the missing digital CH model for our sites, we used the in-situ bulk CH as a substitute. With the in-situ bulk CH data we can test the effect of CH on the model results, but cannot provide spatial predictions in form of maps. Hence, models using CH (PS4 - PS6) were excluded from spatial predictions.

### 2.3.4 Input data for model development

We used data from FE and RB plots to train and test (internally validate) the regression models (n = 82 for DM; n = 81 for N). As REM data was not acquired at the EL site, field data from the EL plots (n = 32) was excluded from the model training. However, the field data from the EL plots was used as an additional external validation of the models utilizing data from the SEQ sensor (Fig. 2) brown part; see chapter 2.3.5.

### 2.3.5 Model evaluation procedure

To derive robust statistics, the regression models were built using a 6-fold cross-validation and repeated ten times with random data splits. Each repetition is connoted as 'iteration' throughout the manuscript. For each iteration, the data is again randomly split into 6 folds; 5 folds to train a model and the hold-out fold to test the model. The corresponding cross-validated evaluation metrics are denoted with a subscript "*cv*". The model evaluation metrics used in the study are the averages from the test folds of the ten iterations. Ground observations from the EL site were used to validate the models based on SEQ data without further site-specific training – for this site REM data was unavailable (Fig. 2; corresponding evaluation metrics indexed with a subscript "*val*"). Evaluation metrics used are coefficient of determination of the validation ($R^2$), root mean square error (RMSE), relative RMSE (rRMSE), and bias (Bias) (Eq. 1 – 4). All metrics were averaged over the 10 iterations.

$$R^2 = 1 - \frac{\sum(y_i - \hat{y_i})^2}{\sum(y_i - \bar{y})^2} \tag{1}$$

$$RMSE = \sqrt{\frac{\sum(\hat{y_i} - y_i)^2}{n}} \tag{2}$$

$$rRMSE = \frac{RMSE}{y_{max} - y_{min}} \tag{3}$$

$$Bias = \frac{\sum(\hat{y_i} - y_i)}{n} \tag{4}$$

where $y$ is an observed value, $\hat{y}$ is a prediction, and $n$ is the number of samples. Relative RMSE is normalized by the observed data range and used to compare regression models with unequal data input following Richter et al. (2012).

### 2.3.6 Model implementation

We used GNU R 3.6 (R Core Team, 2021) for model implementation. GBM was built using the R package "gbm3" (Greenwell et al., 2020) and RF via the R package "randomForest" (Breiman, 2001). Linear regression models were built using all available predictors (LM_full) and the best subset of predictors (LM_best) using variable selection. The variable selection was done by an exhaustive search, i.e. evaluate the Akaike Information Criterion (AIC) (Akaike, 1973) of all possible combinations via *regsubsets* function in the R package "leaps" (Lumley, 2020). Interactions among the predictors were considered in the ML models but not explicitly in the linear models using interaction terms. We did not include interaction terms in the LMs, as the linear models with (first- and second-orders) interaction yielded very large prediction errors in the cross-validation scheme (results from the preliminary analysis, not shown here).

### 2.3.7 Variable importance

In ML, measuring variable importance (Strobl, 2008) is a standard way to evaluate an overall impact of a specific predictor, often among a large number of highly-correlated predictors. In this study, we evaluated variable importance to see how the different predictors overall contribute to the model performance. It is our interests to identify if there is a small number of dominant predictors, or rather a combination of many predictors that contain the crucial information. We investigated the variable importance (VarImp) of the predictors used in the ML regression models in each data and model combination.

For each model, we collected variable importance measures from each six-fold and averaged them, which was repeated in the 10 iterations. As each iteration yielded unequal model performance, the importance metrics of each iteration was normalized by $R^2$ of each iteration before averaging, which resulted in the mean VarImp and its uncertainty range. Note that variable importance measures are based on reduction of Mean Squared Error (MSE), but calculated differently for each ML algorithm. For GBM, we used the relative influence measure suggested in Friedman (2001) and, for RF, permutation based out-of-bag importance of Breiman (2001). Various R packages were used to calculate variable importance, depending on the algorithm (Greenwell et al., 2020; Lumley, 2020; Meinshausen, 2017, 2006; R Core Team, 2021).

### 2.3.8 Mappings: Spatial predictions of DM and N concentration

Spatial predictions were calculated for models that do not need CH data (i.e. PS1 – PS3). We used the models to predict DM and N values for the entire UAS scenes of the three sites. The models are with 10-iteration, thus predicted 10 times, and averages and coefficient of variations are reported. Note that spatial plant trait estimates may be only valid for un-shaded and vegetated grassland pixels.

### 2.3.9 Statistical tests for the marginal model performance

Model performance metrics were averaged over sensors, model algorithms, and predictor sets to derive marginal performance with respect to each component. We used non-parametric statistical methods to test the differences in $R^2$ and RMSE. For sensors and algorithms ($n_{treat} = 2$), we used the non-parametric Wilcoxon signed rank test (Wilcoxon, 1945). For predictor sets ($n_{treat} = 4$), we used the non-parametric Kruskal-Wallis rank sum test (Kruskal and Wallis, 1952) to test overall effect and Dunn's rank sum test (Dunn, 1964) to carry out post-hoc tests between treatments. We used R packages 'stats' and 'dunn.test' (Dinno, 2017; R Core Team, 2021).

## 3. Results

### 3.1 Variable interdependencies

Correlations between variables measured in the field can affect the modelling of DM and N concentration or can even be exploited to improve the modelling. We created scatterplots of selected variables (Fig. 3) and calculated the Spearman

correlation coefficient of canopy height and DM or N concentration, respectively (Fig. 3a, b). Canopy height values varied

between 3 and 21 cm (median = 10 cm, n = 116), and were significantly correlated with DM (r = 0.69, p-value < 0.01), but not

with N concentration (r = 0.02, p-value > 0.1). We also found no statistically significant correlation between DM and N

concentration (r = 0.12, p-value > 0.1; Fig. 3c). Based on these results, we would expect that canopy height could improve the

modelling of DM, but not of N concentration, and that any successful modelling of N concentration does not simply reflect a

correlation of the spectral data with DM.

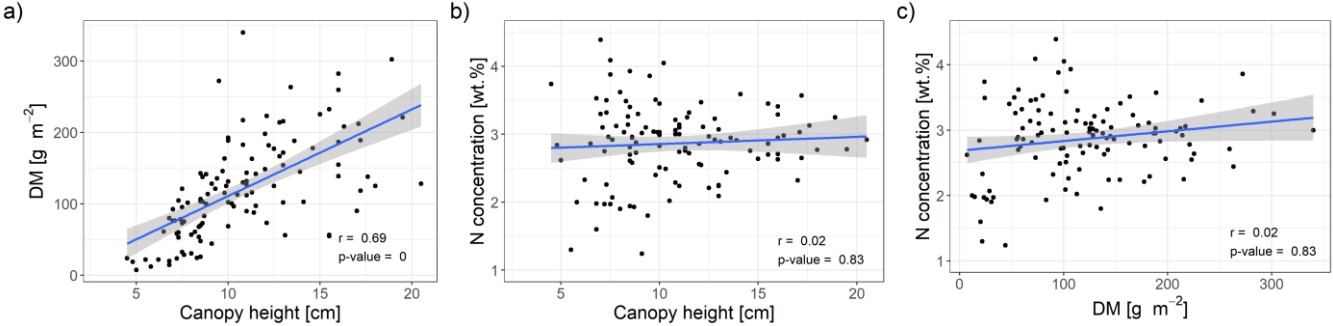

**Figure 3.** Scatter plots of field measurements with linear regression line. a) Canopy height vs. DM; b) Canopy height vs. plant N concentration; c) DM vs. plant N concentration. Spearman correlation coefficients and corresponding p-values are indicated in the figures. The shaded area corresponds to the standard error bounds of the fitted linear regression line.

### 3.2 Biophysical and spectral characteristics of field samples

The spectral discrimination of grasslands samples with different levels of DM or N concentration is a prerequisite for the

estimation of DM and N concentration with multispectral data. In our study, the DM values of the measured subplots varied

between 7 and 340 g m$^{-2}$ (median = 113 g m$^{-2}$, n = 114), and plant N concentration between 1.2 and 4.4 wt.% (median = 2.9

wt.%, n = 113). Despite we targeted homogenous grassland plots, there was a distinct spatial within-plot variability

(Schucknecht et al., 2020a), which however can be reflected by the spatial resolution of the UAS-based multispectral data. In

general, the spectral profiles of selected subplots (Fig. 4) follow the expected shape of vegetated surfaces with low reflectance

values in the visible range of the spectrum and higher reflectance values in the NIR region. Subplots with different DM and N

concentration values show slightly different spectral profiles with highest standard deviations of the reflectance values in the

red edge and NIR band. However, the spectral profiles of subplots with different DM or N concentration values do not follow

a clear pattern, e.g. with monotonically increasing reflectance of the NIR band with increasing DM. There is a positive linear

relationship between NIR reflectance and DM, but this is not very strong (Fig. 5). Additionally, these spectral profiles have

altered patterns for the two sensors (Fig. 4), with the SEQ sensor generally showing higher reflectance values (Fig. SF1).

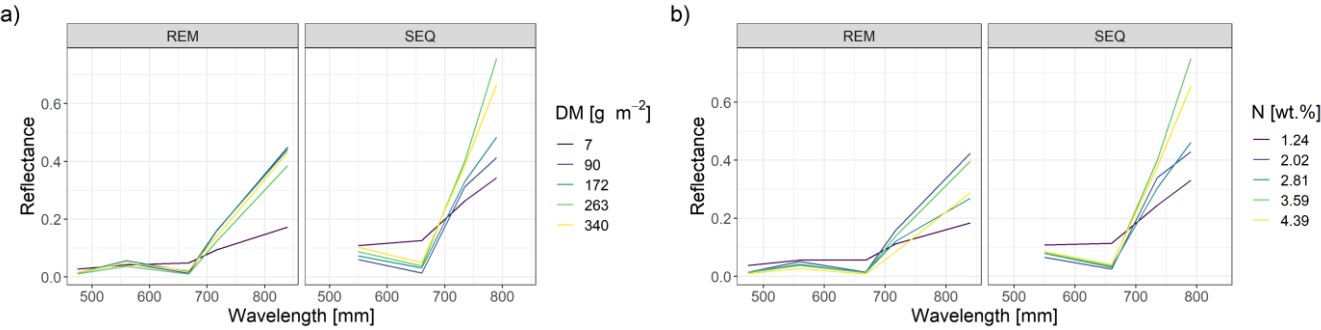

**Figure 4.** Spectral profiles of subplots with different levels of DM (a) and N concentration values (b). Shown are the profiles of the subplot samples with min and max values as well as the ones that have DM or N concentration values that approximately correspond to the 25th, 50th and 75th percentile.

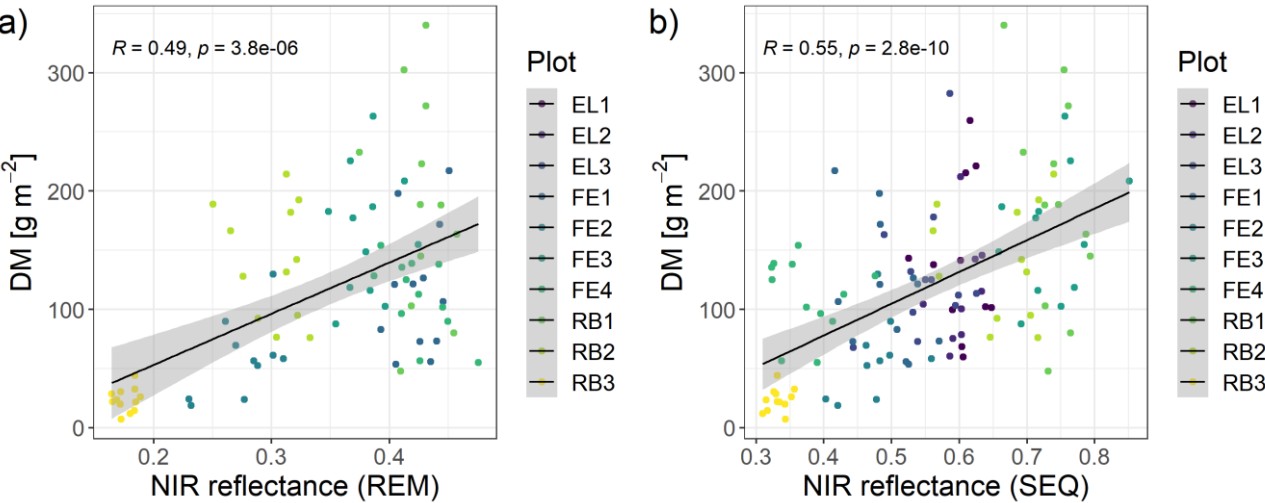

**Figure 5.** Scatterplots of NIR reflectance vs. DM with linear model fit (Spearman correlation coefficient and p-value indicated in the plot). a) for REM sensor; b) for SEQ sensor. Note: there are less data points for REM as there were no flights with this sensor at the Eschenlohe site.

### 3.3 Hyper-parameter calibration

During the calibration process, the model performance increased as improved parameter sets are used in the course of the iteration procedure (Supplementary Fig. SF2). Compared to initial model performance, which is based on randomly sampled 12 parameter sets from the given ranges (i.e. first 12 time steps in calibration (Bischl et al., 2014)), the magnitude of improvement on average was 11%. The difference between the lowest error achieved and the initial error is 19.4% (GBM) and 5.5% (RF) in DM estimation, and 16.1% (GBM) and 2.9% (RF) in N concentration estimation, averaged for the REM and SEQ sensor. Thereby the best error (i.e., the smallest error achieved by $i_{th}$ iteration) did not substantially decrease as the optimizer approaches the global optimum after 400 iterations in most of the cases (Supplementary Fig. SF2). However, in some cases better parameter values were still discovered at the very end of the iteration procedure (e.g., PS1 in Supplementary

Fig. 1a). RF parameters changed less than GBM parameters along the iterations (Supplementary Fig. 1b and d). Furthermore,
parameter proposals are less fluctuating for RF than for GBM as shown in the distance between consecutive parameter
proposals (Supplementary Fig. SF3).

### 3.4 Model results

Our results indicate that the ML algorithms performed substantially better than the linear models in estimating DM and plant
N concentration (Table 4, Table 5, Supplementary Table ST3). ML algorithms yield an average regression performance of
420 0.44 for $R^2_{cv}$. Throughout the sensor-predictor set combinations, average (avg) $R^2_{cv}$ was 0.48 for DM (rRMSE$_{cv,avg}$ = 15.9%;
Table 4) and 0.40 for plant N concentration (rRMSE$_{cv, avg}$ = 15.2%; Table 5). In contrast, the `baseline' linear models are very
low in $R^2_{cv}$, seemingly unsuitable to estimate DM and plant N concentration (all models $R^2_{cv} \leq 0.1$; Supplementary Table ST3).
Therefore, we focus in the following on the detailed results from the ML models (Fig. 5, Fig. 6, Table 4, Table 5) and further
investigate their characteristics.

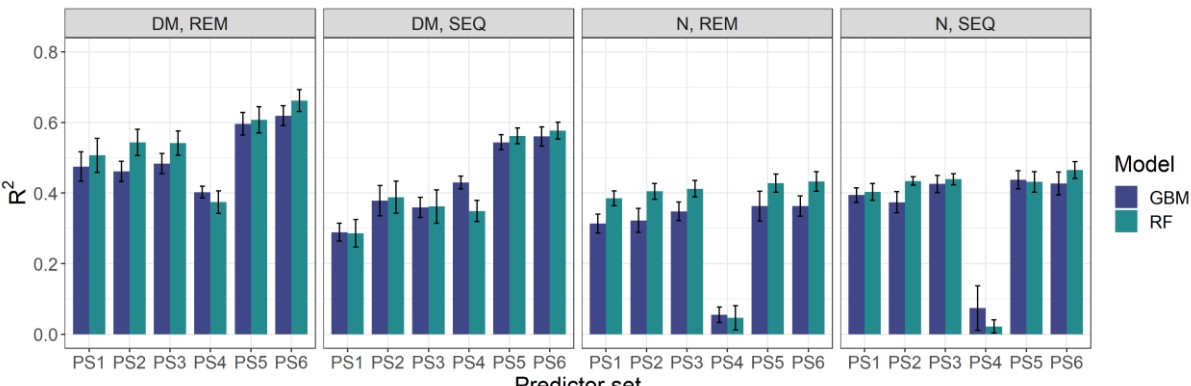

**Figure 6.** Overview of modelling results for all tested combinations of parameters (DM, plant N concentration), sensors (REM, SEQ), ML
algorithms (GBM, RF), and predictor sets (PS1: raw reflectance data; PS2: VI; PS3: raw reflectance data + VI; PS4: canopy height; PS5:
raw reflectance data + canopy height; PS6: raw reflectance data + VI + canopy height). The bars show the mean of the cross-validated
coefficient of determination ($R^2$) and the error bars represent ± standard deviation of the 10 iterations per model.

The optimal combination of sensors, predictor sets and ML algorithm leads to a notable increase in model performance
compared to the average performance of all ML models – for both DM and plant N concentration (Table 4, Table 5, Fig. 5).
The best model for the estimation of DM ($R^2_{cv}$ = 0.67, rRMSE$_{cv}$ = 12.6%) is a RF model that utilizes all possible predictors
(PS6) with REM sensor data (Fig. 6a). The best model for plant N concentration ($R^2_{cv}$ = 0.47, rRMSE$_{cv}$ = 14.2%) is achieved
by the combination of RF, the PS6 predictor set and SEQ input data (Fig. 6d).

The bias of our tested ML models varies between -2.5 and 2.2 g m$^{-2}$ for DM (bias$_{cv, avg}$ = 0.1 g m$^{-2}$) and between -0.04 and 0.01
wt.% for N concentration (bias$_{cv, avg}$ = 0.00 wt.%). Although overall biases are low (bias$_{cv, avg}$ = < 1% of the mean DM and mean
N observation), the models tend to underestimate high DM and high N plant concentration (Fig. 6, Supplementary Fig. SF7
and SF8).

**3.4.1 Effect of different sensors**

The effect of the multispectral sensor on the model performance varied by the different grassland parameters. For DM, the models that built on REM input data perform better ($R^2_{cv, avg}$ = 0.52, $RMSE_{cv}$ = 50.6 g m$^{-2}$) than models that built on SEQ data ($R^2_{cv, avg}$ = 0.39, $RMSE_{cv}$ = 57.5 g m$^{-2}$) averaged over algorithms and predictor sets (Table 4). This difference is statistically significant for $R^2_{cv}$ and $RMSE_{cv}$ (p-value < 0.01; Supplementary Fig. SF4a and SF4c). Additionally, we tested the models built on REM data without the blue band, to see if the performance gap is due to the additional blue band of REM. The results show generally better performance of the REM-without-blue-band-based models ($R^2_{cv, avg}$ = 0.54) than SEQ-based models (Supplementary Table ST3), suggesting that the better performance of models using REM data is not (entirely) related to the additional blue band.

Considering the estimation of plant N concentration (Table 5), models utilizing SEQ input data perform generally better ($R^2_{cv, avg}$ = 0.43, $RMSE_{cv}$ = 0.50 wt.%) than those using REM input data ($R^2_{cv, avg}$ = 0.32, $RMSE_{cv}$ = 0.52 wt.%) and these differences are statistically significant for $R^2_{cv}$, but not for $RMSE_{cv}$ (p-value < 0.01 for $R^2_{cv}$, p-value = 0.092 for $RMSE_{cv}$; Supplementary Fig. SF4b and SF4d).

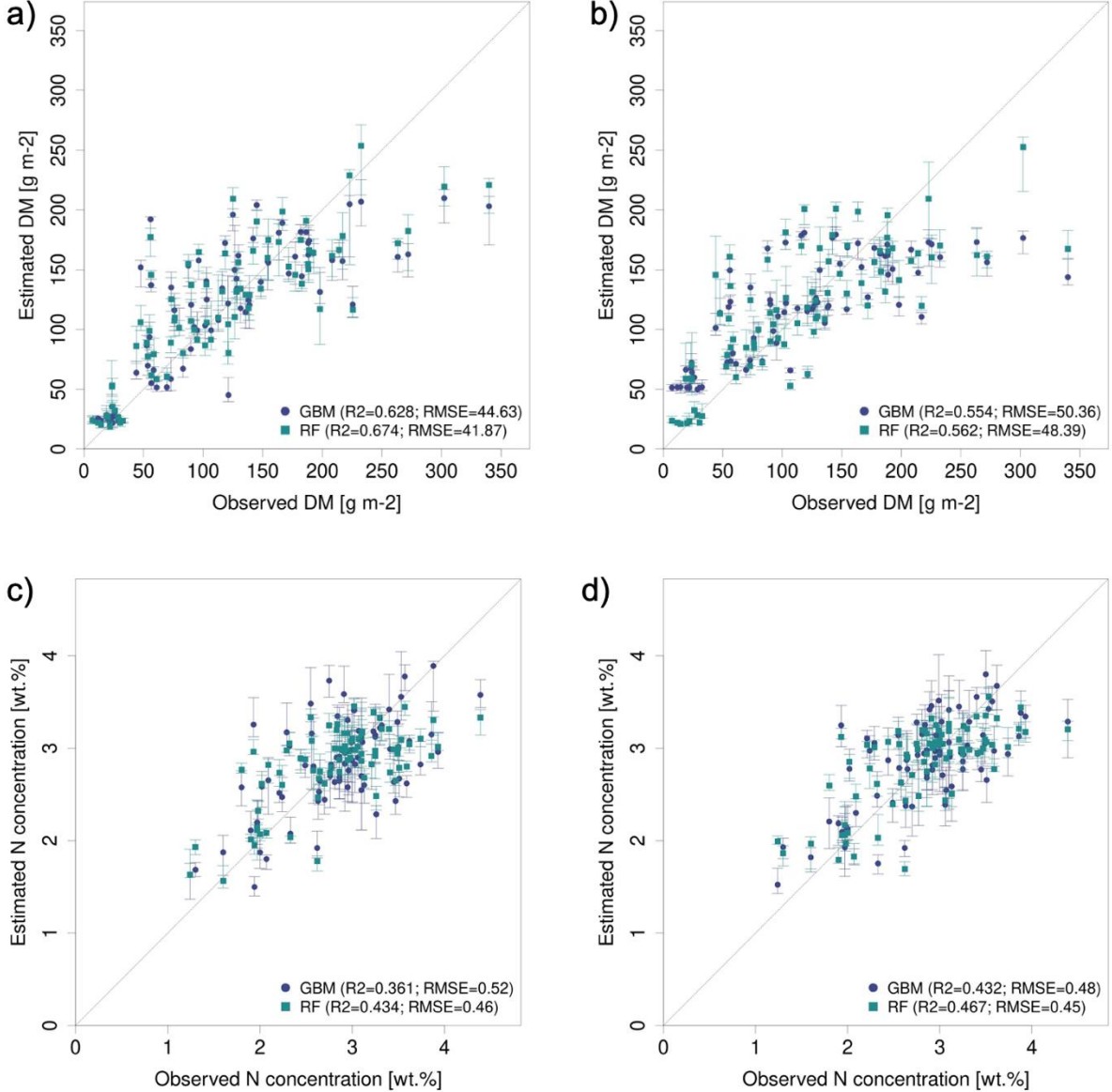

**Figure 7.** Observed vs. estimated parameter values for the best performing predictor set PS6 (raw reflectance data + VI + canopy height) comparing GBM and RF models. a) for DM with REM data, b) for DM with SEQ data, c) for N concentration with REM data, d) for N concentration with SEQ data. The error bars reflect 90% prediction intervals, defined by 5th and 95th percentiles of the 10 iterations.

**Table 4.** Overview of the DM models and cross-validation evaluation metrics for all combinations of sensors (REM, SEQ), predictor sets (PS1: raw reflectance data; PS2: VI; PS3: raw reflectance data + VI; PS4: canopy height; PS5: raw reflectance data + canopy height; PS6: raw reflectance data + VI + canopy height), and ML algorithms (GBM, RF). The unit of $RMSE_{cv}$ and absolute $bias_{cv}$ is g m$^{-2}$. All metric values of single sensor-predictors-algorithm combinations are averages of the 10 iterations. Best results per sensor in bold. The first eleven rows per parameter show aggregated median results (e.g. median of all DM models). $N_{obs} = 82$.

| Parameter | Sensor | Predictors | Model | $R^2_{cv}$ | $RMSE_{cv}$ | $rRMSE_{cv}$ | $Bias_{cv}$ |
|---|---|---|---|---|---|---|---|
| | all | all | all | 0.48 | 53.0 | 15.9 | 0.10 |
| | REM | all | all | 0.52 | 50.6 | 15.2 | 0.34 |
| | SEQ | all | all | 0.39 | 57.5 | 17.3 | 0.00 |
| | all | all | GBM | 0.47 | 53.3 | 16.0 | 0.13 |
| | all | all | RF | 0.52 | 50.6 | 15.2 | 0.09 |
| | all | PS1 | all | 0.38 | 57.6 | 17.3 | 0.25 |
| | all | PS2 | all | 0.43 | 55.0 | 16.5 | -0.37 |
| | all | PS3 | all | 0.43 | 55.5 | 16.7 | -0.28 |
| | all | PS4 | all | 0.39 | 55.0 | 16.5 | -0.06 |
| | all | PS5 | all | 0.58 | 48.0 | 14.4 | 0.92 |
| | all | PS6 | all | 0.59 | 46.5 | 14.0 | 0.14 |
| | | PS1 | GBM | 0.47 | 53.9 | 16.2 | 0.34 |
| | | | RF | 0.50 | 51.8 | 15.6 | 1.89 |
| | | PS2 | GBM | 0.47 | 53.4 | 16.0 | -0.85 |
| | | | RF | 0.54 | 49.5 | 14.9 | 0.94 |
| | | PS3 | GBM | 0.49 | 52.6 | 15.8 | -0.96 |
| | REM | | RF | 0.55 | 49.2 | 14.8 | 0.34 |
| | | PS4 | GBM | 0.40 | 53.3 | 16.0 | -0.07 |
| | | | RF | 0.38 | 55.1 | 16.6 | -0.09 |
| | | PS5 | GBM | 0.59 | 47.8 | 14.4 | 2.07 |
| DM | | | RF | 0.61 | 46.0 | 13.8 | 1.45 |
| | | PS6 | GBM | 0.63 | 44.6 | 13.4 | 0.10 |
| | | | RF | **0.67** | **41.9** | **12.6** | 2.19 |
| | | PS1 | GBM | 0.30 | 61.3 | 18.4 | 0.16 |
| | | | RF | 0.30 | 61.8 | 18.6 | -0.97 |
| | | PS2 | GBM | 0.38 | 59.0 | 17.7 | -2.48 |
| | | | RF | 0.40 | 56.7 | 17.0 | 0.10 |
| | | PS3 | GBM | 0.36 | 58.3 | 17.5 | -0.48 |
| | SEQ | | RF | 0.37 | 58.4 | 17.6 | -0.09 |
| | | PS4 | GBM | 0.44 | 54.8 | 16.5 | 0.19 |
| | | | RF | 0.35 | 60.7 | 18.2 | -0.06 |
| | | PS5 | GBM | 0.54 | 49.5 | 14.9 | 0.40 |
| | | | RF | **0.56** | **48.2** | **14.5** | -0.09 |
| | | PS6 | GBM | 0.55 | 50.4 | 15.1 | 0.19 |
| | | | RF | **0.56** | 48.4 | **14.5** | 0.07 |

**Table 5.** Overview of the plant N concentration models and cross-validation evaluation metrics for all combinations of sensors (REM, SEQ), predictor sets (PS1: raw reflectance data; PS2: VI; PS3: raw reflectance data + VI; PS4: canopy height; PS5: raw reflectance data + canopy height; PS6: raw reflectance data + VI + canopy height), and ML algorithms (GBM, RF). The unit of $RMSE_{cv}$ and absolute $bias_{cv}$ is wt.%. All metric values of single sensor-predictors-algorithm combinations are averages of the 10 iterations. Best results per sensor in bold. The first eleven rows per parameter show aggregated median results (e.g. median of all DM models). $N_{obs} = 81$.

| Parameter | Sensor | Predictors | Model | $R^2_{cv}$ | $RMSE_{cv}$ | $rRMSE_{cv}$ | $Bias_{cv}$ |
|---|---|---|---|---|---|---|---|
| | all | all | all | 0.40 | 0.48 | 15.2 | 0.00 |
| | REM | all | all | 0.36 | 0.51 | 16.3 | 0.00 |
| | SEQ | all | all | 0.43 | 0.48 | 15.1 | 0.00 |
| | all | all | GBM | 0.36 | 0.51 | 16.3 | 0.00 |
| | all | all | RF | 0.42 | 0.47 | 14.9 | 0.00 |
| | all | PS1 | all | 0.39 | 0.48 | 15.3 | 0.01 |
| | all | PS2 | all | 0.39 | 0.49 | 15.6 | 0.00 |
| | all | PS3 | all | 0.42 | 0.48 | 15.1 | 0.00 |
| | all | PS4 | all | 0.04 | 0.66 | 21.1 | 0.00 |
| | all | PS5 | all | 0.43 | 0.46 | 14.7 | 0.00 |
| | all | PS6 | all | 0.43 | 0.47 | 15.0 | -0.01 |
| | | PS1 | GBM | 0.31 | 0.51 | 16.3 | 0.00 |
| | | | RF | 0.38 | 0.49 | 15.4 | 0.01 |
| | | PS2 | GBM | 0.31 | 0.54 | 17.3 | -0.02 |
| | | | RF | 0.40 | 0.48 | 15.2 | -0.01 |
| | | PS3 | GBM | 0.34 | 0.51 | 16.3 | -0.03 |
| | REM | | RF | 0.41 | 0.47 | 15.0 | 0.00 |
| N | | PS4 | GBM | 0.05 | 0.57 | 18.2 | 0.00 |
| | | | RF | 0.03 | 0.71 | 22.5 | 0.00 |
| | | PS5 | GBM | 0.36 | 0.52 | 16.6 | -0.01 |
| | | | RF | **0.43** | 0.47 | 14.8 | 0.00 |
| | | PS6 | GBM | 0.36 | 0.52 | 16.4 | -0.04 |
| | | | RF | **0.43** | **0.46** | **14.7** | -0.01 |
| | | PS1 | GBM | 0.39 | 0.48 | 15.2 | 0.01 |
| | | | RF | 0.40 | 0.47 | 15.0 | 0.01 |
| | | PS2 | GBM | 0.38 | 0.50 | 15.9 | 0.00 |
| | | | RF | 0.43 | 0.46 | 14.8 | 0.00 |
| | | PS3 | GBM | 0.42 | 0.48 | 15.2 | 0.00 |
| | SEQ | | RF | 0.44 | 0.46 | 14.7 | 0.00 |
| | | PS4 | GBM | 0.05 | 0.62 | 19.7 | 0.00 |
| | | | RF | 0.02 | 0.72 | 22.9 | 0.00 |
| | | PS5 | GBM | 0.44 | 0.46 | 14.6 | 0.01 |
| | | | RF | 0.43 | 0.46 | 14.7 | 0.00 |
| | | PS6 | GBM | 0.43 | 0.48 | 15.2 | -0.01 |
| | | | RF | **0.47** | **0.45** | **14.2** | 0.00 |

## 3.4.2 Effect of predictor sets and variable importance

The selection of the subsets of predictors clearly influences the performance of DM models (Fig. 5, Fig. SF6, Fig. SF7, Table 4, Table AT1). For REM-based models, all predictor sets that only use spectral data (i.e. PS1: raw reflectance; PS2: VI; PS3: raw reflectance + VI) show a similar and slightly higher performance than the predictor set using only canopy height (PS4).

For SEQ-based models, the use of VI (PS2, PS3) or canopy height only (PS4) improves the model performance compared to the baseline scenario just using raw reflectance data (PS1). The best model results for REM- and SEQ-based models are obtained by combining spectral data with canopy height information (PS5, PS6). These predictor sets show significantly higher $R^2_{cv}$ and lower $RMSE_{cv}$ than predictor sets with spectral data (PS1, PS2, PS3) or canopy height information only (PS4).

For plant N concentration (Fig. 6, Fig. SF6, Fig. SF8, Table AT1), the models using spectral predictors (PS1, PS2, PS3) show a similar model performance ($R^2_{cv, avg} = 0.39 – 0.42$) that is insignificantly lower than for models using all available predictors (PS6; $R^2_{cv, avg} = 0.43$). Models using just canopy height as predictor are not capable to predict N concentration ($R^2_{cv, avg} = 0.04$) and perform notably worse than all other predictor sets (p-value < 0.05 for $RMSE_{cv}$; p-value < 0.05 for $R^2_{cv}$ except for comparison of PS1 and PS4).

**Variable importance**

The analysis of variable importance shows some general observations (see Supplementary Table ST4 for detailed results). The added value of the inclusion of structural information in the predictor set for the estimation of DM is further substantiated by canopy height having the highest relative variable importance in all predictor sets where it is included (PS5, PS6) independent of the used sensor data and ML algorithm. For example, canopy height accounts on average for 39% of error reduction, measured by relative importance for all models using PS5, and 28% for the all models using PS6. Besides, the NIR band shows the highest variable importance for DM estimation (Table ST4) over all sensor-algorithm combinations in the baseline scenario (PS1). For predictor set PS5 (raw reflectance + CH), the NIR band has the second highest variable importance after canopy height, again over all sensor-algorithm combinations. If VI and raw reflectance were included in the predictor set (PS3, PS6), one (PS3 with SEQ-GBM combination) or a few VI (all other sensor-algorithm combinations) are ranked higher than the raw reflectance band with the highest variable importance. Vegetation indices that have a variable importance of at least 5% in all sensor-algorithm combinations of predictor sets including VI (PS2, PS3, PS4) are Datts, $NDVI_{re}$, and RR1, which all include the NIR band in their formula (Table ST1).

In contrast to DM, there is in general no clear order or dominance of a certain predictor recognizable over all sensor-algorithm combinations for the estimation of N concentration. Canopy height shows a much lower variable importance compared to the DM models, with average relative importance of 15% for PS5 and 8% for PS6.

**3.4.3 Effect of modelling algorithm**

Overall, the two tested ML algorithms show a smaller difference in model performance than the two sensors and the predictor sets (Table 4, Table 5, Supplementary Fig. SF5) for DM and plant N concentration. RF usually performs better (DM: $R^2_{cv, avg}$ = 0.52; N: $R^2_{cv, avg}$ = 0.42) than GBM (DM: $R^2_{cv, avg}$ = 0.47; N: $R^2_{cv, avg}$ = 0.36), but the difference in $R^2_{cv}$ and $RMSE_{cv}$ between GBM and RF is not significant for DM as well as the difference in $RMSE_{cv}$ for N concentration (p-value > 0.1). Models using REM data generally show a higher difference in model performance between GBM and RF, both when considering DM and N.

Noticeable is the distribution of relative importance of the predictors between the two ML algorithms (Supplementary Table ST4). GBM is often characterised by one dominating variable (especially for DM), which has substantially higher relative importance than other variables. In contrast, RF models show a more gradual decrease in variable importance for the subsequent ranked predictors.

### 3.4.4 External model validation with data from Eschenlohe (EL)

The models based on SEQ sensor data were additionally validated against the ground observations from the EL site that were not used for model training. Considering DM models (Table 6), the validation results for the EL plots show lower $R^2$ and higher RMSE values compared with the cross-validated model results of the RB and FE sites (Table 4). Furthermore, the model predictions for EL are more biased ($bias_{val, avg}$ = -15.4 g m$^{-2}$). As seen in the cross-validated results, particularly high DM values are generally not well captured with a clear downward bias (Fig. 7, Supplementary Fig. SF9). The best model for

the estimation of DM in EL ($R^2_{val}$ = 0.51, $RMSE_{val}$ = 41.0 g m$^{-2}$, $rRMSE_{val}$ = 18.4%) uses RF with predictor set PS6 (raw reflectance data + VI + canopy height). Prediction of DM for EL is significantly improved by the use/inclusion of canopy height as predictor (PS4, PS5) and to a lesser extent by VI (PS2, PS3) (Table 6, Supplementary Fig. SF9).

All N concentration models show very low $R^2$ values ($R^2_{val} \leq 0.03$) for the external validation site (Table 6). The models for the external validation site predict levels of N concentration > 2.2 wt.%, but do not sufficiently capture the variations between

520 2.2 and 4.1 wt.% (Fig. 8b, Fig. SF10). The GBM model using PS4 (canopy height) predicts a single value for all N observations (Fig. 8b) implying that the model is not sensitive in this range. The levels of rRMSE ($rRMSE_{val, avg}$ = 28.0%) are also higher than those of the cross-validated results ($rRMSE_{cv, avg}$ = 16.3%).

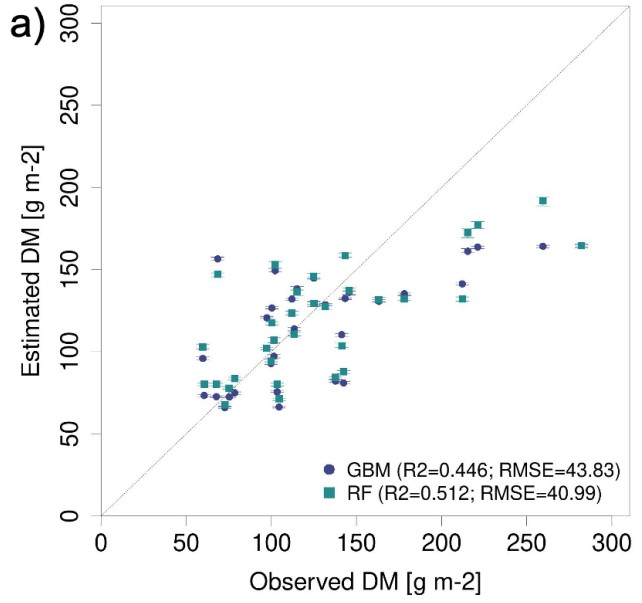
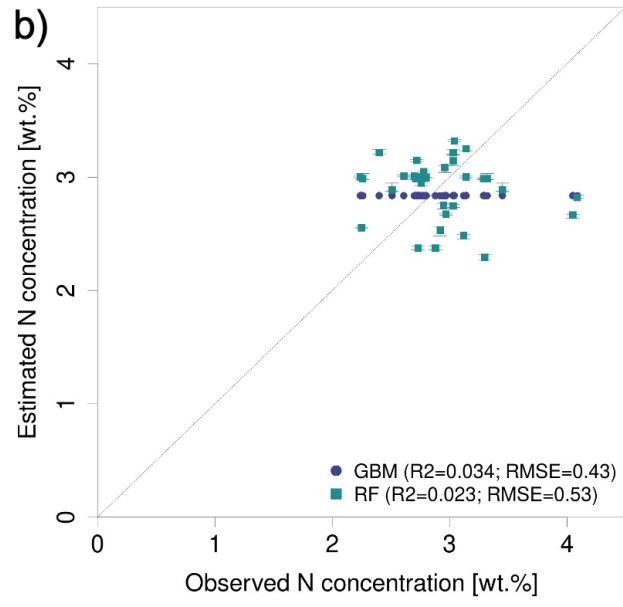

**Figure 8.** Observed vs. estimated DM (left) and N concentration (right) in the external site EL using SEQ data with the best performing predictor set: (a) DM models based on PS6 (raw reflectance data + VI + canopy height) and (b) N concentration models based on PS4 (canopy height)

### 3.5 Spatial predictions

The spatial DM prediction with the best performing spatial model (RF model with REM data and predictor set PS3 – raw reflectance data + VI) for the RB-North site shows within-field variability (Fig. 9a). Furthermore, the extensively managed field around plot RB3 is characterized by very low DM values, which corresponds to field observations. The individual iterations of this model combination show very similar DM predictions that just differ slightly in areas with low DM as indicated by the coefficient of variation (CV) of the 10 iterations (Fig. 9b). Compared to this, the difference in spatial DM prediction between different DM model combinations (Fig. 9c) is much higher with highest differences (CV > 30%) occurring at places with very low DM. The main spatial pattern between different model combinations are similar, but less pronounced spatial pattern may differ depending on the used combination of sensor, ML algorithm and predictor set.

The spatial prediction of plant N concentration for the RB-North site (Fig. 9e) also show a certain within-field variability and the extensively managed field around plot RB3 stands out with very low N concentrations. Most of the grassland pixels outside the extensive field are characterized by N concentrations between 2.5 wt.% and 3.5 wt.%. The differences between the ten iterations of one model combination (Fig. 9e) and between different model combinations (Fig. 9g) are generally lower than for the DM models.

The spatial prediction maps for the other sites (Supplementary Fig. SF11 to SF14) also indicate within-field and between-field variability of DM and plant N concentration as well as highest differences between models at grassland areas with low values of DM and N concentration.

**Table 6.** External model validation with EL site using SEQ sensor data. The unit of $RMSE_{val}$ and $bias_{val}$ of DM is g m$^{-2}$ and wt.% for plant N concentration. Shown are all combinations of predictor sets (PS1: raw reflectance data; PS2: VI; PS3: raw reflectance data + VI; PS4: canopy height; PS5: raw reflectance data + canopy height; PS6: raw reflectance data + VI + canopy height) and ML algorithms (GBM, RF). All metric values of single predictors-algorithm combinations are averages of the 10 iterations. Best results in bold. The first nine rows per parameter show aggregated median results (e.g. median of all DM models). $N_{obs} = 32$.

| Parameter | Predictors | Model | $R^2_{val}$ | $RMSE_{val}$ | $rRMSE_{val}$ | $Bias_{val}$ |
|---|---|---|---|---|---|---|
| | all | all | 0.27 | 51.8 | 23.3 | -15.4 |
| | all | GBM | 0.24 | 52.7 | 23.7 | -15.4 |
| | all | RF | 0.27 | 51.0 | 22.9 | -15.0 |
| | PS1 | all | 0.03 | 57.3 | 25.8 | -16.2 |
| | PS2 | all | 0.21 | 54.7 | 24.6 | -16.6 |
| | PS3 | all | 0.22 | 51.9 | 23.3 | -16.0 |
| | PS4 | all | 0.29 | 52.8 | 23.8 | -15.3 |
| | PS5 | all | 0.42 | 45.3 | 20.3 | -14.6 |
| | PS6 | all | 0.48 | 42.4 | 19.1 | -12.1 |
| | PS1 | GBM | 0.01 | 57.9 | 26.0 | -15.8 |
| DM | | RF | 0.06 | 56.7 | 25.5 | -16.7 |
| | PS2 | GBM | 0.20 | 58.3 | 26.2 | -18.9 |
| | | RF | 0.23 | 51.1 | 23.0 | -14.2 |
| | PS3 | GBM | 0.19 | 52.9 | 23.8 | -15.9 |
| | | RF | 0.25 | 50.9 | 22.9 | -16.0 |
| | PS4 | GBM | 0.29 | 52.5 | 23.6 | -14.8 |
| | | RF | 0.30 | 53.2 | 23.9 | -15.8 |
| | PS5 | GBM | 0.41 | 45.3 | 20.3 | -15.1 |
| | | RF | 0.42 | 45.3 | 20.3 | -14.1 |
| | PS6 | GBM | 0.45 | 43.8 | 19.7 | -13.3 |
| | | RF | **0.51** | **41.0** | **18.4** | **-10.9** |
| | all | all | 0.02 | 0.51 | 27.4 | 0.2 |
| | all | GBM | 0.01 | 0.52 | 28.1 | 0.3 |
| | all | RF | 0.02 | 0.51 | 27.4 | 0.2 |
| | PS1 | all | 0.01 | 0.48 | 26.2 | 0.2 |
| | PS2 | all | 0.02 | 0.57 | 31.1 | 0.3 |
| | PS3 | all | 0.02 | 0.54 | 29.2 | 0.3 |
| | PS4 | all | 0.03 | 0.48 | 26.2 | -0.1 |
| | PS5 | all | 0.02 | 0.47 | 25.3 | 0.2 |
| | PS6 | all | 0.00 | 0.56 | 30.3 | 0.3 |
| | PS1 | GBM | 0.01 | 0.47 | 25.5 | 0.16 |
| N | | RF | 0.02 | 0.50 | 26.9 | 0.22 |
| | PS2 | GBM | 0.02 | 0.62 | 33.5 | 0.41 |
| | | RF | 0.01 | 0.53 | 28.6 | 0.27 |
| | PS3 | GBM | 0.00 | 0.57 | 30.6 | 0.35 |
| | | RF | **0.03** | 0.51 | 27.8 | 0.24 |
| | PS4 | GBM | **0.03** | **0.43** | **23.4** | -0.10 |
| | | RF | 0.02 | 0.53 | 28.9 | **-0.05** |
| | PS5 | GBM | **0.03** | 0.46 | 25.0 | 0.15 |
| | | RF | 0.00 | 0.47 | 25.6 | 0.19 |
| | PS6 | GBM | 0.01 | 0.62 | 33.4 | 0.36 |
| | | RF | 0.00 | 0.50 | 27.1 | 0.24 |

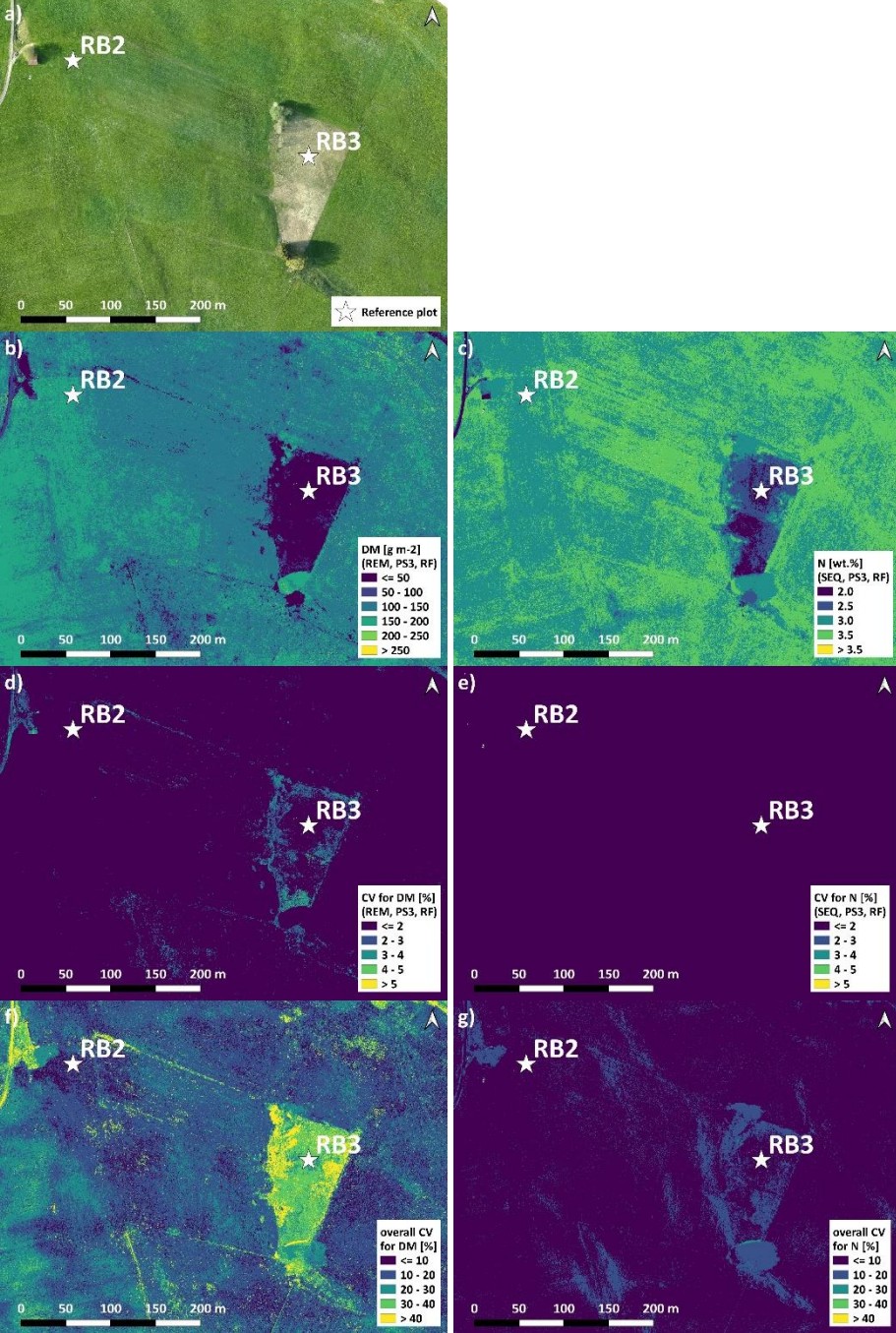

**Figure 9.** Spatial estimation at the RB-North site. a) orthophoto for comparison, b) DM with REM-PS3-RF-combination (best spatial prediction), c) N concentration with SEQ-PS3-RF-combination (best spatial prediction), d) CV of DM with REM-PS3-RF-combination, e), e) CV of N concentration with SEQ-PS3-RF-combination, f) overall CV of DM for all PS1 and PS3 models, g) overall CV of N concentration for all PS1 and PS3 models. Predictor set PS3: raw reflectance data + VI. Estimation of DM and N concentration represent

the mean of the 10 iterations for the selected model. Note that spatial estimates are only valid for un-shaded and vegetated grassland pixels.

## 4. Discussion

In this study, we analysed the potential to estimate DM and plant N concentration with low-cost UAS-based data in pre-Alpine managed grasslands. We tested two multispectral sensors, three statistical models and six different predictor sets and evaluated marginal effects of them. The models were trained and validated with in-situ data. An emphasis was put on the calibration of the two ML algorithms GBM and RF.

### 4.1 Suitability of multispectral data to estimate DM and plant N concentration

The spectral differences between samples of different DM and plant N concentration levels (Fig. 4) indicate that an estimation of these two grasslands parameters could be obtained by multispectral UAS data. However, the link between spectral pattern and the level of DM or plant N concentration, respectively does not seem to be straight-forward as e.g. demonstrated by the week linear correlation between DM and NIR reflectance and the unsuitability of linear models to estimate DM and plant N concentration. Potential reasons for the rather week bivariate relationship could be the different species compositions of the subplots, differences during the acquisition (time during the day, clouds), as well as the radiometric correction of the multispectral sensors (see chapter 4.7 for more detailed discussion of UAS acquisition issues), which can be challenging and a source of uncertainty (Olsson et al., 2021). Accordingly, our model results confirmed that the estimation of DM and plant N concentration is feasible when applying machine learning algorithms, but with a noticeable error range. The best models using all available multispectral data (i.e. raw reflectance and VI) plus bulk canopy height information achieved a $R^2_{cv}$ of 0.67 ($rRMSE_{cv}$ = 12.6%) for DM and a $R^2_{cv}$ of 0.47 ($rRMSE_{cv}$ = 14.2%) for N concentration. These findings are in line with other studies which also confirmed the suitability of ML algorithms for grassland parameter estimation based on UAS data. The multi-temporal study of Grüner et al. (2020) on an experimental farm with legume-grass mixtures applied also a RF model to the spectral reflectance data and VI of a SEQ sensor and achieved a $R^2_{cv}$ of 0.62 and $rRMSE_{cv}$ of 17% for DM estimation. The authors showed that the modelling performance was clearly improved by adding texture parameters to the predictor set ($R^2_{cv}$ = 0.87 and $rRMSE_{cv}$ of 11%). The analyses of Wijesingha et al. (2020) addressed the prediction of forage quality in grasslands with multi-temporal hyperspectral UAS-data in the wavelength range between 450 – 998 mm. They compared different regression algorithms and found that support vector regression worked best for the prediction of crude protein ($R^2_{cv}$ = 0.81, cross-validated normalised $RMSE_{cv}$ = 9.6%), but RF yielded similarly good results.

In our study, plant N concentrations models did not perform as good as the DM models, generally achieving much lower accuracies. The same decrease in accuracy was also found for the external validation site, where the N models marked much lower $R^2$ and higher RMSE values compared to the DM models. Furthermore, the N concentration models benefitted to a much lesser extent from the addition of the canopy height information to the spectral predictors. One reason for the less good performance of N concentration models in our study is certainly the result of the lower value range of plant N concentration

(coefficient of variation of all samples, $CV_{N, all} = 19.8\%$) compared with DM ($CV_{DM, all} = 57.3\%$). Most of the N concentration in leaves is related to pigments like chlorophyll and proteins involved in photosynthesis with the most important being Rubisco (Ollinger, 2011 and references therein). While pigments are the dominant absorbers in the visible range of the electro-magnetic spectrum, non-pigment compounds mainly have absorption features at longer wavelengths (Ollinger, 2011 and references therein). In his review, Ollinger (2011) summarizes several hyperspectral vegetation indices that are used for chlorophyll detection. However, the author emphasised that the effects of plant N concentration on leaf spectra are still unclear e.g. if spectral reflectance is mainly driven by direct effects of N-containing compounds or indirect effects of related traits. In a recent publication, Berger et al. (2020) question the use of the commonly used chlorophyll-nitrogen relationship as it is not maintained after the vegetative growth stage and propose to quantify instead leaf protein concentration. The authors recommend the use of hyperspectral sensors for N quantification as the spectral signatures related to proteins are subtle and mainly located in the shortwave infrared (SWIR) region. This should be further explored together with ML models trained on radiative transfer models (Berger et al., 2020) or other modelling approaches like crop, plant growth or biogeochemical models assimilating remote sensing data.

### 4.2 Importance of machine learning algorithms and their calibration

Linear models generally failed to capture variability both in DM and plant N concentration estimation as expressed in the cross-validated predictive performance. In all cases, the predictive performance was substantially better for ML algorithms. Linear models do not cope well with a large number of highly-correlated predictors as well as with non-linearity (Marchese Robinson et al., 2017). Explorative modelling techniques such as manual feature engineering in linear models, including advanced models such as generalized linear models may help achieving better model performance to the level of ML algorithms, as those methods can cover some weaknesses of LM.

As shown in the results, hyper-parameter calibration of ML algorithm confirmed to be crucial, leading to 11% improvement in model performance. The lowest error was not revealed in the early iterations and the parameter searching is discovering lower error values almost until the end of the iterations, suggesting that there is a risk of drawing inference based on sub-optimal result when `default' parameter values are used. Note that the impact of calibration was more pronounced for GBM, in contrast the calibration of RF was seemingly more efficient (i.e. discovered optimal parameter values in a smaller number of iterations) in line with previous studies (Bernard et al., 2009; Probst et al., 2019). The MBO algorithm used in the calibration is more efficient in exploring a high dimensional parameter space than naive searching algorithms such as grid searching, random sampling, or Latin hypercube sampling (Bischl et al., 2014). MBO does not need binning, or discretisation and can also stop earlier than scheduled unlike grid searching, when it reaches the prescribed goal, or yields no improvement. Those features enables MBO to explore parameter domains more comprehensively and effectively. A caveat of the adaptive algorithm is that it needs prescribed stopping rules (e.g., number of iterations or % of improvement), though such stopping rules do not assure the optimal performance. Objective stopping rules should be further investigated in future applications such as metrics

based on convergence [e.g. Gelman-Rubin diagnostic commonly used in Markov Chain Monte Carlo modelling (Brooks and Gelman, 1998; Gelman and Rubin, 1992)].

In this study, a nested cross-validation scheme is applied ensuring that the calibration routine does not see the hold-out data. Otherwise, the calibration could rather lead to loss of predictive power (Vabalas et al., 2019). We should note that the DM and plant N concentration values of the validation site (EL) are within the range of observations made in the training sites (Schucknecht et al., 2020a). Training data spanning a wide range of observed DM and N concentration values, and maybe originating from different types of grassland at different times in the growing season, would be desired to build a generally

applicable model. Overall, the two tested ML algorithms yielded comparable model performance after calibration; RF performed slightly better (higher $R^2_{cv}$ and lower $RMSE_{cv}$), but without statistical significance.

## 4.3 Impact of different sensors

The two tested multispectral sensors affected the model performance in different ways depending on the considered grassland parameter. While REM-based models outperformed SEQ-based models in the estimation of DM, SEQ-based models yielded

significantly better results for plant N concentration estimation in terms of $R^2_{cv}$ and $RMSE_{cv}$. The two sensors have a different spectral setup with slightly different central wavelengths. While SEQ has a wider green and red band, REM has an additional blue band. Furthermore, the red edge bands of the two sensors are not overlapping as well as the NIR bands. Considering some reflectance spectra of grass (e.g. USGS spectral characteristics viewer: https://landsat.usgs.gov/spectral-characteristics-viewer; Rossi et al. 2020, Rossini et al. 2012), we would assume that the difference in the red edge band position could lead to

significant reflectance differences between SEQ and REM for one sample (due to steep increase of reflectance in the red edge region). The effect of the different central wavelength in the NIR band might be less pronounced (as the NIR plateau is reached) and the different band width of the green and red band are expected to have a negligible effect on the difference in reflectance value between the two sensors. These and other constructional differences in the sensors might partly explain differences in the spectral profiles of certain subplots (Fig. 4) and thus differences in model performance.

A possible reason for SEQ sensor showing generally higher reflectance values than the REM sensor with sometimes even implausibly high reflectance values in the NIR band (> 0.7; Fig. 5, Fig. SF1) might be calibration issues of the sensor. The radiometric correction of the multispectral sensors, which is needed to convert digital numbers to surface reflectance, can be challenging and a source of uncertainty (Olsson et al., 2021), which might be even more important than the spectral specification of the sensors. Poncet et al. (2019) compared different radiometric correction methods for the Parrot Sequoia

sensor including the manufacturer method using a one-point calibration plus a sunshine sensor like in our study. The authors found no method allowing to maximize data accuracy for all bands and different flight conditions. The manufacturer-recommended method that includes the sunshine sensor yielded comparable data accuracy as the best empirical method, but could be improved by the combination with an empirical calibration (Poncet et al., 2019). In their study Olsson et al. (2021) evaluated the accuracy of the Parrot Sequoia camera and sunshine sensor, highlighting the influence of the camera temperature

on the sensitivity of the camera, the influence of the atmosphere on the images as well as the influence of the orientation of

the sunshine sensor on raw irradiance data. We were not aware that the Sequoia sensor needs to be sufficiently warm before reaching a stable sensitivity (Olsson et al., 2021). Since we took images of the sensor-specific calibration targets before each flight, this might have negatively influenced our radiometric calibration and introduced uncertainty in the reflectance data. The above mentioned difference between the reflectance levels measured by the two sensors is not likely to have a considerable

effect on the results of the estimation of canopy traits, as only data from one of the sensors is used in model fitting. However, this means that the results from the model fit will only be valid for the reflectance data from the sensor used and parameters from different model fits based on data from the two sensors are thus not directly comparable. Proper calibration of the two sensors would therefore be advised in future studies to make the results more generally applicable and comparable.

Besides constructional and radiometric correction aspects, changing illumination conditions may have contributed to
differences in the reflectance values of a certain subplot in the comparison of the two sensors. Data acquisition at the Fendt site was partly affected by passing clouds, which is also visible in the data of the irradiance sensor of the SEQ (not shown). In general, it would be beneficial to have in-situ reflectance data of the subplots (e.g., from a field spectroradiometer) to validate the reflectance values of the two UAS sensors, but these are not available.

In summary, we could not conclusively clarify the exact reasons that led to the differences in spectral signatures of the two
sensors and their model performance. This would require additional laboratory and field tests that are out of the scope of this study. However, our study indicates that the REM sensor might be preferred for applications targeting biomass estimation in pre-Alpine grasslands.

## 4.4 Impact of different predictor sets

The models solely dependent on UAS data (PS1, PS2, PS3) were moderately good, both for DM and plant N concentration
estimation. Adapting vegetation indices is straightforward and feasible under any condition. Therefore, it is important to evaluate its added value. With regard to the cross-validation, models using PS2 (VI) and PS3 (raw reflectance + VI) showed generally higher $R^2_{cv}$ and lower $RMSE_{cv}$ values than that of PS1 (raw reflectance), but the difference was not significant neither for DM nor for N concentration. The addition of VI seemed to be more important for the external validation as it significantly increased the predictive performance of DM for the validation site EL compared to the baseline scenario PS1 (Supplementary
Fig. SF7). The addition of VI in N models for the validation on the EL did not improve the model performance.

This finding is in line with other studies, which also pointed out that VI and other arithmetic band combinations may help to improve the prediction accuracy for vegetation related quantitative and thematic variables (Maschler et al., 2018; Seo et al., 2016). The benefits are observed despite the fact that VI do not really add "new" information, which is not yet contained in the spectral signatures (Baret and Guyot, 1991; Atzberger et al., 2011). The empirically observed benefits are most probably
linked to the reduction of shadow-related brightness effects.

Our results highlight the importance of combining spectral information with canopy height for the estimation of DM. Models using spectral and CH information (PS5, PS6) had significantly higher $R^2_{cv}$ and lower $RMSE_{cv}$ values than those using just spectral information (PS1, PS2, PS3) or just CH information (PS4). The effect of combining canopy height with spectral data

in the predictor set is larger than the effect of the used ML algorithm or sensor. It may suggest that canopy height is playing a crucial role as it reflects seasonal growth and canopy structure independent from the spectral information.

Canopy height as sole predictor was not suitable to estimate plant N concentration. The effect of adding CH to spectral data in the predictor set was slightly positive, but not significant. This low relevance of CH in the estimation of plant N concentration could be expected from the missing correlation between N concentration and canopy height (Fig. 4).

High-resolution UAS-based RGB data can be used to derive canopy height models that can then be integrated in the spatial DM estimation. Some studies already utilized canopy height information in the estimation of grassland yield (Grüner et al., 2019; Lussem et al., 2020, 2019; Viljanen et al., 2018). However, it needs to be kept in mind that a precise and high-resolution DTM is required to derive reliable vegetation structure estimates from UAS imagery (Poley and McDermid, 2020). The generation of such high-quality DTMs can be challenging in areas with a dense vegetation canopy as it is the case for our pre-Alpine grasslands. In their review, Poley and McDermid (2020) reported different methods for DTM generation that have been applied, when ground points were not well visible: active sensors like LiDAR and terrestrial or aerial laser scanning as well as terrain interpolation based on high-accuracy GPS data collected on the ground (see references in Poley and McDermid, 2020). A low-cost alternative to the active sensors might be a UAS-based digital surface model of the freshly cut grassland as used for example in Lussem et al. (2019).

In addition to CH, texture and the spatial variation of the image elements, was shown to correlate with vegetation structure and heterogeneity (Gallardo-Cruz et al., 2012) and can vary with the phenological stage of the vegetation (Culbert et al., 2009). Grüner et al. (2020) demonstrated that the modelling performance of DM in legume-grass mixtures was improved by the addition of texture parameters in the predictor set.

### 4.5 Spatial predictions

Spatial pattern in DM and N concentration can differ depending on the used combination of sensor, ML algorithm and predictor set of the model, especially with respect to less strong pattern. The magnitude of these differences in terms of the coefficient of variation of all used models is larger for DM than for N concentration with highest CV in areas of low DM or N concentration values. However, without additional spatial information (e.g. on soil properties, soil moisture, species composition, etc.) it is hard to interpret these differences in spatial pattern and assess the quality in spatial prediction of the single models. We plan to adapt the developed ML models for multiple campaign applications and already collected field data of several plots at different dates during the growing season 2019 and 2020. In this context, it would be an interesting research question if the spatial pattern of single models are persistent in time.

### 4.6 Transferability of model results

A limitation of this study is that the model is trained using data of a single flight campaign at each site, which may raise the question about the transferability of the developed models, i.e. do the relationships apply also for data from other sites and dates (across different growth stages). The training data was collected from several sampling sites differing in management,

species composition, current canopy height and phenological stage to increase the general validity of the results of a single campaign. The spatial transferability of the developed models was (partly) tested with the external validation with data from the Eschenlohe site that was not used in model building. The results indicate that DM models work moderately well at other sites that are within the value range of the training sites (Fig. 8a). With respect to the estimation of N concentration, the models

failed to predict the variability of N at the validation site (Fig. 8b). However, the model is fairly good in capturing the mean N concentration values (e.g., small bias in Fig. 8b), implying it needs to learn more about the low and high N domains. Therefore, we expect that both DM and N models would benefit from an increased training data base that capture a wider range of values and originate from different grasslands.

The applicability of the developed models across different phenological stages is partly accounted for by the use of training

data originating from different phenological phases (Table 1). However, the full range of phenological phases is not covered. Including training data from multiple sampling campaigns over the growing season would be desired for further studies, as Rossini et al. (2012) showed a change in the reflectance spectra of grasslands for different times in the year.

Another aspect of transferability is whether the trained model is reusable in other problem domains. The models we used (GBM and RF) are not directly resuable in other problem domains, meaning the trained weights are not usable if there is any

change in model set-up (e.g. addition or removal of predictors, changing response variables). This is due to the fact that the two algorithms belong to the family of 'shallow' learning algorithms, in contrast to 'deep' algorithms. They learn features in a small number of layers (i.e., shallow), compared to deep algorithms often comprised of many several layers. However, the shallow algorithms are still useful to other studies in the sense that model diagnosis metrics (i.e. variable importance) and optimal model structure (i.e. calibrated parameter values) are informative to similar research questions. These metrics are

useful to understand the processes and help design field campaigns and build new models in another domain, space, and time. In constrast to deep algorithms, they are relatively straightforward to build and train, while moderately well and robust in capturing variations.

### 4.7 Challenges of UAS studies with low-cost sensors

**Acquisition of UAS data**

The acquisition of UAS data has some advantages over satellite imagery like the flexibility in flight conduction (no fixed overflight day and time) and the possibility to acquire data during cloud cover. Having the full control of flight scheduling also allows to decide, whether the acquisition conditions are sufficient for the specific application and the corresponding data quality requirements or whether the campaign needs to be postponed. On the other hand, the UAS flight campaigns depend on good weather conditions (no precipitation, few wind) and it is sometimes not easy to find a suitable date, where the weather

fits and all people from the field campaign team have time. In general, stable illumination conditions during the flights are desirable to avoid negative effects on the data quality (e.g., Assmann et al. 2018). In practice, one often has to face the trade-off between data availability and data quality. From our experience, it can sometimes be difficult to find an optimal date for conducting a UAS campaign in the desired phase of the grassland development stage. Therefore, we also have to accept

changing illumination conditions (e.g., due to passing clouds, different sun angle) in order to have at least an acquisition even

if the data quality might not be optimal. The present study represents such a real-world case, where we were searching for a good weather window and where we had to coordinate quite a lot of people from different institutions. Finally, we had sub-optimal illumiation conditions at one of our sampling sites due to passing clouds.

In addition, the duration of UAS data acquisition at the first sampling day was quite long as we have flown two sites and the field team was not yet practiced. Here, we identified a clear potential for optimisation. Measuring the position of the GCPs

and the centres of the subplots with the GNSS in "topopoint mode" (a measurement just takes a few seconds, but it requires a mobile phone coverage) instead of static mode, would save a lot of time. Furthermore, the workflows in the field could be improved to require less time. With these optimisations the duration of the flights could be shortened and conducted around solar noon. However, the tradeoff between number of flights (i.e. number of different sites covered) and data quality aspects due to varying sun angle partly remains.

**Data quality**

Issues with the quality of low-cost UAS sensors and radiometric calibration have been reported in the literature (e.g., Aasen et al. 2018; Assmann et al. 2018; Olsson et al., 2021; Poncet et al., 2019). In our study, the difference in the spectral profiles of subplots between the two used multispectral sensors raises questions related to the quality of the obtained data, but could not finally be addressed. The placement of spectral reflectance targets in the overflight area (Aasen et al. 2018; Assmann et al.

2018) and the simultaneous collection of field spectrometer measurement would allow for a better assessment of the quality of the obtained UAS data. The practical feasibility of the latter option might be a constraint, especially in view of the required field personal and duration of field work. The lack of a standard quality control information layer provided by the data processing workflow of Pix4D as compared to certain satellite data products, is a drawback for the user.

In summary, we think that there are quite some measures to improve the quality of UAS data, but not all can be considered at

775 all times in practical applications. At the end the user of UAS data needs to accept that the quality cannot be as good as for satellite imagery and should consider this aspect in the interpretation of the derived products. However, extending campaign periods and increasing the replication and the sampling frequency could actually lead to a reduction of uncertainty especially under limited budget (Kim et al., 2022). Thus we may well advocate the use of low-cost sensors in a range of applications, which require high-spatial resolution and flexible application options.

Until today, low-cost UAS is the only affordable way to acquire individual-level spatial information for a specific location and time. In precision farming, such fine-grained spatial information support the optimization of fertilizer application, weed and disease management, harvest, and irrigation (e.g., Tsouros et al. 2019). For such applications, the value of low-cost sensors are rather high even if their spectral quality is not at the level of satellite or high-precision sensors. Spatial patterns acquired from a low-cost sensor product can be directly used to derive spatial gradients, and as a complement to satellite products.

## 5. Conclusions

Spatially explicit information on grassland biomass and quality could improve local farm management and support regional-scale assessments, e.g. on nitrogen cycling. This study aimed to develop, assess, and apply models to estimate DM and plant N concentration of pre-Alpine grasslands on the field-scale with UAS-based multispectral data and canopy height information. We tested two different sensors, three statistical modelling approaches and six input data sets with respect to their effect on model performance using in-situ data from ten permanent grasslands. Our results indicate that ML algorithms are able to estimate DM and plant N concentration, whereby DM models showed better performance in terms of $R^2$ and RMSE. The combined use of spectral and canopy height information in the predictor set significantly improved the prediction for DM, but not plant N concentration. Including VI was also beneficial for DM prediction, but to a lesser extent. Data from REM sensor yielded significantly better model performance results for DM estimation, while SEQ data was significantly better for plant N concentration estimation. Overall, machine learning algorithms utilizing UAS-based multispectral data and canopy height information proved to be a promising tool for the estimation of DM and plant N concentration in pre-Alpine grasslands. Further research should address the transferability of approaches, e.g. by extending the calibration and validation data base, the improvement of the models, e.g. by incorporation of texture parameters, and the spatial up-scaling through the utilization of satellite data.

**Table AT1.** Results of the non-parametric statistical tests between parameter pairs on $R^2_{cv}$ and $RMSE_{cv}$. Three different tests were carried out: Wilcoxon-test for sensors and algorithms ($N_{treat} = 2$), Kruskal-Wallis-test for predictor sets overall, Dunn's test between predictor sets (see details in Section 2.3.5). Note that all the tests were done for paired samples. Symbols for significance level: ** ($p \leq 0.01$), * ($p \leq 0.05$), - ($p > 0.05$).

| | Tested parameter pairs | p-value ($R^2_{cv}$) | | p-value ($RMSE_{cv}$) | Significance |
|---|---|---|---|---|---|
| | Sensors | 0.001 | ** | 0.000 | ** |
| | Algorithms | 0.151 | - | 0.266 | - |
| DM | Overall | 0.007 | ** | 0.011 | * |
| | PS1-PS2 | 0.258 | - | 0.309 | - |
| | PS1-PS3 | 0.345 | - | 0.274 | - |
| | PS1-PS4 | 0.480 | - | 0.421 | - |
| | PS1-PS5 | 0.006 | ** | 0.005 | ** |
| | PS1-PS6 | 0.003 | ** | 0.004 | ** |
| | PS2-PS3 | 0.401 | - | 0.460 | - |
| | PS2-PS4 | 0.242 | - | 0.382 | - |
| | Predictor sets PS2-PS5 | 0.032 | * | 0.018 | * |
| | PS2-PS6 | 0.016 | * | 0.014 | * |
| | PS3-PS4 | 0.326 | - | 0.345 | - |
| | PS3-PS5 | 0.018 | * | 0.023 | * |
| | PS3-PS6 | 0.008 | ** | 0.018 | * |
| | PS4-PS5 | 0.005 | ** | 0.008 | ** |
| | PS4-PS6 | 0.002 | ** | 0.006 | ** |
| | PS5-PS6 | 0.382 | - | 0.460 | - |
| | Sensors | 0.003 | ** | 0.092 | - |
| | Algorithms | 0.016 | * | 0.233 | - |
| N | Overall | 0.029 | * | 0.042 | * |
| | PS1-PS2 | 0.309 | - | 0.480 | - |
| | PS1-PS3 | 0.159 | - | 0.212 | - |
| | PS1-PS4 | 0.061 | - | 0.029 | * |
| | PS1-PS5 | 0.097 | - | 0.159 | - |
| | PS1-PS6 | 0.074 | - | 0.227 | - |
| | PS2-PS3 | 0.309 | - | 0.198 | - |
| | Predictor sets PS2-PS4 | 0.020 | * | 0.032 | * |
| | PS2-PS5 | 0.212 | - | 0.147 | - |
| | PS2-PS6 | 0.171 | - | 0.212 | - |
| | PS3-PS4 | 0.005 | ** | 0.004 | ** |
| | PS3-PS5 | 0.382 | - | 0.421 | - |
| | PS3-PS6 | 0.326 | - | 0.480 | - |
| | PS4-PS5 | 0.002 | ** | 0.002 | ** |
| | PS4-PS6 | 0.001 | ** | 0.004 | ** |
| | PS5-PS6 | 0.440 | - | 0.401 | - |

**Code availability.** The codes used in the preparation of this paper are available upon request from the authors.

**Data availability.** The field data set used in this study is available on the PANGAEA repository at https://doi.org/10.1594/ PANGAEA.920600 (Schucknecht et al., 2020b).

**Supplement.** The supplement related to this article is available online at:

**Author Contributions**: A.S.: conceptualization, data curation, formal analysis, investigation, methodology, project administration, writing – original draft, visualization; B.S.: data curation, formal analysis, investigation, methodology, software, validation, visualization, writing – original draft; A.K.: formal analysis, funding acquisition, investigation, resources, writing – review and editing; SA: methodology, writing – original draft; CA: writing – review & editing; RK: funding acquisition, investigation, project administration, writing – review & editing.

**Competing interests:** The authors declare that they have no conflict of interest.

**Funding:** This research was conducted within the SUSALPS project (https://www.susalps.de/). The work was supported by the German Federal Ministry of Education and Research (grant numbers 031B0027A, 031B0516A, 031B0027E, 031B0516E, 031B0516F). We acknowledge support by the KIT-Publication Fund of the Karlsruhe Institute of Technology.

**Acknowledgments:** We thank our colleagues from KIT, IMK-IFU for their great support during the field and laboratory work, our partners from TU Munich for the N analysis, Kerstin Grant from Agricultural Centre Baden-Wuerttemberg (LAZBW) for the determination of the phenological stages of the plots, SAPOS for providing correction data, and the local farmers for their cooperation.

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
