# Peer review of "Estimating dry biomass and plant nitrogen concentration in pre-Alpine grasslands with low-cost UAS-borne multispectral data – a comparison of sensors, algorithms, and predictor sets"

_Biogeosciences, 2021_

## Author Comment (AC1)

**Response to referee #1 comment**

Referee comment on "Estimating dry biomass and plant nitrogen concentration in pre-Alpine grasslands with low-cost UAS-borne multispectral data – a comparison of sensors, algorithms, and predictor sets" by Anne Schucknecht et al., Biogeosciences Discuss., https://doi.org/10.5194/bg-2021-250-RC1, 2021

We thank referee #1 for the constructive comments. Please find below, how we want to address the raised issues (referee comments in italic) in a revised version of the manuscript.

**Main issue 1 (regarding single date acquisition of UAS and field data)**

*The study is entirely based on a single field campaign, therefore during one specific stage of grasslands phenological development (presumably early season). This is an important limitation which is not discussed. I would recommend to more explicitly highlight this and discuss the implications.*

We are aware of the limitation of a single field campaign. The study presented in the manuscript tests the general approach of grassland trait estimation with an application of immediate grassland trait mapping. To increase the general validity or the results of a mono-temporal campaign, we selected several sites differing in species composition, current growth height and nutrient status, which allowed us to compile a dataset of variable grassland traits. We will include this aspect in the material and method section and add a paragraph in the discussion chapter to address the applicability of the study and requirements for other phenological stages (multi-temporal applications).

**Main issue 2 (regarding limitations of UAS)**

*The study in the introduction suggests that UAS can bring significant advantages as compared to satellite or airborne data in mountain regions. However, I would argue they have also significant limitations, especially when low-cost sensors are used. By looking at the study results, I have the impression (perhaps wrong) that issues related to data acquisition and quality (e.g. long acquisition time, calibration and incident radiation measurement) might play an important role in explaining the relatively poor model performances. However, this is not mentioned or discussed in a clear way (there are some points, but not really a section discussing the challenges in UAS data acquisition and quality). I would find valuable to see some more discussions on the issues that might be related to UAS use (even better if supported by some analyses).*

We agree with the referee that studies utilizing UAS data have their own challenges and drawbacks. Therefore, we will include a section in the discussion chapter that will discuss the challenges with UAS and low-cost sensors in general and the limitations of our study in particular (including the point raised in issue 1). Anyhow, we assume that the relative differences between sensors and predictor sets etc. are still valid and honest outcome of this study, even though there could be noise due to the UAS data acquisition.

**Minor issues**

*Introduction:*

*There is a bit of mix between Alpine/pre Alpine etc, while most statements are valid for both. Perhaps if the area falls within the Alpine space (geographic region) there is no need to specify Pre. Alpine is sufficient and more details about the sites are given in the methods.*

We prefer to keep the distinction between Alpine and pre-Alpine to highlight that our study covers the hilly Alpine foreland and not the high-elevation pastures, which would be associated with Alpine. Anyhow, we will adapt some sentences to make this clearer.

*Perhaps 'often long' is unnecessary*

"Often long" will be removed in the revised manuscript.

*The sentence is unclear*

Unfortunately, we are not sure to which sentence this comment refers. Could you please specify it?

*86-89 This might be truth for specific cases, but it is important to keep in mind the limitations of sensors technologies onboards UAS.*

Thank you for the comment. We will include a sentence on the limitations of UAS sensors.

*Here and elsewhere it is mentioned canopy height data were not 'available' without explanation. I would suggest avoiding that, as this is explained in the methods. Otherwise, a short justification should be added here too.*

We will follow the suggestion of the referee and adapt the respective sentences accordingly.

**Materials and methods**

*There is no mentioning of the phenological stage of vegetation during the field campaign. This is an important factor.*

We will add the information on the phenological stage by checking the photos of the field campaign.

*9.50 to 16.30 is a quite long interval with expected variations in solar angle and, in mountain regions, shadows and possibly cloudiness. This could be quite a relevant factor affecting the data acquisition.*

We agree with the referee and will include this issue in the discussion chapter (under the new section about challenges of UAS data and their acquisition).

*Is there any indication of the geolocation accuracy?*

The accuracy of the GNSS is provided in line 180ff: The exact coordinates of the GCPs` centres were obtained with a Global Navigation Satellite System (GNSS) receiver (Viva GNSS GS 10, Leica Geosystems AG, Switzerland) run in static mode for 10 minutes which resulted in an accuracy of 0.3 cm in horizontal direction and 0.5 cm in vertical direction in post-processing mode (Datasheet of Leica Viva GNSS GS10 receiver, 2020).

*An area of 3x3 pixels seem very small considering geolocation errors. Assuming the plot should be somehow representative of a wider area, would not be more prudent to have a larger window?*

We selected the 3x3 pixel window to approximately cover the area of the subplot (0.25 m x 0.25 m). Due to the high accuracy (0.003 m) of the GNSS measurements in relation to the window size of ~ 0.25 m, we expect just minor location errors. Another aspect is that although we aimed to sample homogenous plots, we realized in the field (and by further data analysis) that there were small-scale variations due to different plant species. These small-scale variations would be neglected by a larger window size.

**Results**

*It is somewhat surprising the NIR does not follow DM or height, as this should be rather straightforward (unless for very small range). Is there any factor related to the acquisition that might be causing this issue?*

In Figure 4 of the manuscript we just plotted the spectral profiles of a few selected samples (corresponding to min and max values of DM and N as well as the ones that have DM or N concentration values that approximately correspond to the 25th, 50th and 75th percentile). In Figure 4, the NIR reflectance does follow DM, except for the maximum value. We now created additional figures (Figure 1 below) to show the relationship between the reflectance of the NIR band and DM (and canopy height). In fact, there is a positive relationship between NIR and DM, but this relationship

is not the strongest. Therefore, the appearance of Figure 4 depends very much on the selection of the samples.

Potential reasons for the rather week relationship could be the different species compositions of the subplots, differences during the acquisition (time during the day, clouds), as well as the radiometric correction of the multispectral sensors, which can be challenging and a source of uncertainty (Olsson et al., 2021). We will include the new plots and address the issue in the discussion of the revised manuscript version.

[Figure]

*Figure 1. Scatterplots of NIR reflectance vs. DM for the REM sensor (a) and SEQ sensor (b) with linear model fit (Spearman correlation coefficient and p-value indicated in the plot). Note: there are less data points for REM as there were no flights with this sensor at the Eschenlohe site.*

*The doubt of a strong influence of acquisition factors is also supported by the very poor performance of regression as compared to machine learning and the improved performances on DM including VI in the validation. Also the important role of ground canopy height may suggest that as this variable is clearly not affected by the UAS data acquisition. I would suggest to run some tests and eventually add some considerations in the discussions.*

Thanks for raising this issues. As we conceive the issue, in our data the traits are not strongly correlated to spectral signals and VI in bivariate manner. However, it does not only seem to be because of the acquisition error, but other confounding factors involved as well. We will do more diagnosis and discuss the arguments of the reviewer on this topic in the discussion and add relevant literature as evidence.

*The word 'notably' is very often repeated. Sometime is a bit redundant*

We will adapt the text accordingly.

*Figure 7. It is a bit strange to see many points along a line (i.e. same N content) in Fig.7b. Is it correct?*

We have checked the figure and it is correctly reflecting the model outcome. The points along a line means that in this model set-up, GBM does not differentiate most of the samples in predicting N concentration. This is because the domain of the input data values of the test site are new to the algorithm, makes the algorithm to guess it is constant. We will discuss this in the discussion and add potential strategies how to overcome this (i.e. using synthetic sampling and generative modelling).

---

## Author Comment (AC2)

**Response to referee #2 comment**

Referee comment on "Estimating dry biomass and plant nitrogen concentration in pre-Alpine grasslands with low-cost UAS-borne multispectral data – a comparison of sensors, algorithms, and predictor sets" by Anne Schucknecht et al., Biogeosciences Discuss., https://doi.org/10.5194/bg-2021-250-RC2, 2021

We thank referee #2 for the constructive comments. Please find below, how we want to address the raised issues (referee comments in italic) in a revised version of the manuscript.

**Main issue 1 (regarding motivation for comparing two UAS sensors)**

*The motivation for comparing these two UAS sensors is not clear. Are these two types of sensors are popularly used in UAS remote sensing studies? How the findings from the two sensor comparison are relevant to other studies and the UAS remote sensing community? Overall, SEQ and REM sensors are very similar. These two sensors have similar pixel resolution, similar wavelengths in green (550/560 nm), red (660/668 nm), and red edge (735/717 nm). Furthermore, the manuscript pointed out that SEQ performed well for predicting plant nitrogen concentration, while REM had a better performance for predicting dry biomass. However, it is not clear why these two sensors had such different performances in the current manuscript. The analysis and explanation for sensor performance on dry biomass and nitrogen predictions need to be strengthened.*

We agree with the referee that the motivation of comparing two multispectral sensors is not well addressed in the current manuscript version. Several studies applying low-cost UAS sensors for vegetation mapping/monitoring used the Parrot Sequoia sensor and some highlighted certain quality issues (e.g. Olsson et al, 2021; Poncet et al. 2019). We wanted to test if the Micasense RedEdge-M is a good alternative in the low-cost segment and if it has a better performance due to the additional blue band and slightly different central wavelengths/band widths in the other bands. In the revised version of the manuscript we will include some sentences about our motivation to compare two multispectral sensors and strengthen the discussion about the two sensors (also considering the next part of referee comment).

*In Table 2, you labeled 790nm as near infrared. However, we usually refer to 700-800nm as red edge, while wavelengths beyond 800nm as near infrared. From the soil-vegetation radiative transfer modeling view, red edge wavelengths are vital for vegetation chlorophyll content and nitrogen content retrieval. The near infrared is more sensitive to the vegetation canopy structure such as leaf area index and total biomass. From my interpretation, SEQ has two red edge bands and could potentially get better results for nitrogen concentration retrieval, but not dry biomass as lacking information in near infrared. Meanwhile, REM has information on near infrared which is good for biomass retrieval.*

Thanks for the valuable comment. The forth band of the Parrot Sequoia sensor is denoted from the manufacturer as NIR band. It covers the wavelength range of 770 to 810 nm (central wavelength of 790 nm), therefore being in the transition between red edge and NIR according to the referees definition. To foster the comparison with other studies using the Sequoia sensor, we will keep the band names as used by the manufacturer. But we will include a paragraph in the discussion chapter on the differences in the "NIR" band between the Sequoia and RedEdge M sensor and potential implications of it, building on the explanations of the referee and literature information.

**Main issue 2 (regarding motivation for selecting GBM and RF)**

*The motivation for selecting Gradient Boosting Machines and Random Forest is also not clear. Why not other more popular machine learning or statistical approaches, such as partial least-squares regression, LASSO, Ridge, or Neural Networks?*

We share your concern, thanks. Actually, we explored many other algorithms, however the focus of the paper is not comparing a vast number of the ML algorithms, we constrained ourselves to the two

selected algorithms, based on the understanding that those are confirmed to work as good as the others as we cited in Section 2.3.1, Caruana and Niculescu-Mizil, 2006; Fernández-Delgado et al., 2019, 2014; Orzechowski et al., 2018. We will make it clearer in the corresponding section by adding more references about the quality of the chosen algorithms.

*The purpose of applying machine learning algorithms is not only to achieve good model predictive performance. Many machine learning algorithms like random forest can help to identify the relative importance of each feature input. This feature importance analysis is very necessary to understand the relationship between feature inputs and the predicted variables. However, such analysis is missing in this study. I strongly recommend further feature importance analysis to identify scientific linkage among input variables and the predicted variable to strengthen the manuscript result interpretation.*

Thank you for the comment. Actually we diagnosed variable importance using both of the algorithms as mean variable importance (= feature importance) for all tested models is currently provided in the supplementary material (Table ST4) and partly addressed in chapter 3.4.2. (focussing on the most important observations). However, we did not discuss them in depth. Following the referees suggestion, we will provide more information and discussion on the variable importance.

**Main issue 3 (regarding single date acquisition of UAS and field data)**

*The UAS multispectral data were collected from one single flight in each site. How robustness of these results across different growth stages and dates is uncertain?*

We are aware of the limitation of a single field campaign. The study presented in the manuscript tests the general approach of grassland trait estimation with an application of immediate grassland trait mapping. To increase the general validity or the results of a mono-temporal campaign, we selected several sites differing in species composition, current growth height and nutrient status, which allowed us to compile a dataset of variable grassland traits. We will include this aspect in the material and method section and add a paragraph in the discussion chapter to address the applicability of the study and requirements for other phenological stages (multi-temporal applications).

**Main issue 3 (hyper-parameter tuning)**

*Machine learning parameter tunning is a very necessary and common step to implement model training. However, this manuscript highlights the hyper-parameter tunning as one major research question. The innovations of this study need to be strengthened.*

We admit that the importance of ML parameter tuning was pronounced in many of previous studies, but in practice, we observe many ML applications in geo- and environmental-science applications still omitting a comprehensive calibration procedure. In this paper, we want to show that how much progress actually can be achieved in a practical application, for suggesting that this has to be a mandatory step in similar modelling studies (e.g. 10% performance loss). We will clarify this aspect in the discussion section.

**Minor issues**

*There are many abbreviations in Figure 2. The caption should add explanations of these abbreviations for readers.*

We will add explanations of the abbreviations in Figure 2.

*The reflectance values in Figure 4 look quite different from the two sensors. Do you have ground reflectance collection to validate your reflectance?*

Unfortunately, we do not have ground reflectance values for validation. It is important to note that Figure 4 shows the spectral profiles of selected samples for better readability of the plot. An example for all samples is given in the Figure below for NIR reflectance vs. DM (see a comment from referee #1). The figure shows a positive, but not very strong relationship between NIR reflectance and DM as well as the different reflectance value range of the two sensors. Possible reasons for the different

pattern between the REM and SEQ sensor could be the different spectral and radiometric properties, radiometric calibrations and changes in acquisition conditions.

[Figure]

*Figure 1. Scatterplots of NIR reflectance vs. DM for the REM sensor (a) and SEQ sensor (b) with linear model fit (Spearman correlation coefficient and p-value indicated in the plot). Note: there are less data points for REM as there were no flights with this sensor at the Eschenlohe site.*

*The manuscript mentioned that mountain regions have frequent cloud occurrences to argue the weakness of Copernicus satellite missions. However, UAS data collection under cloudy environment also has data quality issues. The manuscript may need to discuss such potential issues and mitigation strategies.*

We agree with the referee that studies utilizing UAS data have their own challenges and drawbacks. Therefore, we will include a section in the discussion chapter that will discuss the challenges with UAS and low-cost sensors in general and the limitations of our study in particular.

*Most parts of the manuscript used nitrogen concentration. However, Figure 6 used nitrogen content in the (c) and (d) subplots.*

We will correct Figure 6.

*The same issue of nitrogen concentration on Figure 7.*

We will correct Figure 7.

*Figure 8 (d) has clear shadows. The reflectance from these shadows needs to be either corrected to real surface reflectance to quantify vegetation traits or simply removed. I don't think the current estimates for areas in tree shadows are right.*

The estimation of DM and N concentration can just be used for vegetated grassland, not affected by shadows. To make this clearer for the reader, we will mention that the plant traits estimates are only valid for un-shaded pixels and vegetated grassland areas, and mask out all non-vegetated and shaded areas in the maps of Figure 8.

*The figure panel design of Figure 8 is strange. We normally put RGB into the first subplot. You have paired maps for DM and N. These paired subplots could be in one row.*

We will rearrange the sub-figures as proposed.

---

## Author Response (AR1)

**Revision notes for referee #1 comment on "Estimating dry biomass and plant nitrogen concentration in pre-Alpine grasslands with low-cost UAS-borne multispectral data – a comparison of sensors, algorithms, and predictor sets" by Anne Schucknecht et al., Biogeosciences Discuss., https://doi.org/10.5194/bg-2021-250-RC1, 2021**

Dear reviewer #1,

Thank you very much for your helpful comments.

You will find our revised manuscript (with track changes) uploaded on the manuscript handling system and below our detailed response to your comments. The original comments from reviewer #1 are in *italic and black* and the responses are in blue. The lines mentioned in the revision note refer to the revised version in track-changes mode of our manuscript.

**Response to reviewer #1**

**Main issue 1 (regarding single date acquisition of UAS and field data)**

*The study is entirely based on a single field campaign, therefore during one specific stage of grasslands phenological development (presumably early season). This is an important limitation which is not discussed. I would recommend to more explicitly highlight this and discuss the implications.*

We are aware of the limitation of a single field campaign. The study presented in the manuscript tests the general approach of grassland trait estimation for the application as immediate grassland trait mapping. To increase the general validity of the results of a single campaign, we selected several sites differing in management, species composition, current canopy height and phenological stage (of the dominant species), which allowed us to compile a dataset of variable grassland traits. We included this aspect in the material and method section (line 145, 156f, Table 1) and added a paragraph in the discussion chapter (line 733 - 742) to address the applicability of the study and requirements for other phenological stages (multiple campaign applications).

**Main issue 2 (regarding limitations of UAS)**

*The study in the introduction suggests that UAS can bring significant advantages as compared to satellite or airborne data in mountain regions. However, I would argue they have also significant limitations, especially when low-cost sensors are used. By looking at the study results, I have the impression (perhaps wrong) that issues related to data acquisition and quality (e.g. long acquisition time, calibration and incident radiation measurement) might play an important role in explaining the relatively poor model performances. However, this is not mentioned or discussed in a clear way (there are some points, but not really a section discussing the challenges in UAS data acquisition and quality). I would find valuable to see some more discussions on the issues that might be related to UAS use (even better if supported by some analyses).*

We agree with the referee that studies utilizing UAS data have their own challenges and drawbacks. Therefore, we included a section in the discussion chapter (chapter 4.7) that discusses the challenges with UAS and low-cost sensors in general and the limitations of our study in particular. Anyhow, we assume that the relative differences between sensors and predictor sets etc. are still valid and honest outcome of this study, even though there could be noise due to the UAS data acquisition.

**Minor issues**

*Introduction:*

*There is a bit of mix between Alpine/pre Alpine etc, while most statements are valid for both. Perhaps if the area falls within the Alpine space (geographic region) there is no need to specify Pre. Alpine is sufficient and more details about the sites are given in the methods.*

We prefer to keep the distinction between Alpine and pre-Alpine to highlight that our study covers the hilly Alpine foreland and not the high-elevation pastures, which would be associated with Alpine. We specified in line 37 what we mean with *pre-Alpine* and *Alpine*. When we write *(pre-)Alpine* in the text, we indicate that both pre-Alpine and Alpine grasslands are addressed.

*Perhaps 'often long' is unnecessary*

Thank you for the comment. We have removed the expression "often long".

*The sentence is unclear.*

After feedback from the editor, we assume the reviewer means the following sentence: "Also the fast data collection and processing and the relatively low cost of many remote sensing data products are advantageous (Wachendorf et al., 2017), as are the often long time series of well calibrated satellite sensors." We reformulated the sentence to clarify the content (line 71-73).

*86-89 This might be truth for specific cases, but it is important to keep in mind the limitations of sensors technologies onboards UAS.*

Thank you for the comment. We added sentences about the limitations of UAS sensors (line 91-96).

*Here and elsewhere it is mentioned canopy height data were not 'available' without explanation. I would suggest avoiding that, as this is explained in the methods. Otherwise, a short justification should be added here too.*

We followed the suggestion of the referee and deleted the respective sentence in the introduction (line 128f).

***Materials and methods***

*There is no mentioning of the phenological stage of vegetation during the field campaign. This is an important factor.*

We added the information on the phenological stage in Table 1 and line 156ff after checking the photos of the field campaign.

*9.50 to 16.30 is a quite long interval with expected variations in solar angle and, in mountain regions, shadows and possibly cloudiness. This could be quite a relevant factor affecting the data acquisition.*

We agree with the referee and included this issue in the discussion chapter (under the new section 4.7 about challenges of UAS data and their acquisition).

*Is there any indication of the geolocation accuracy?*

The accuracy of the GNSS was already provided in the first submitted manuscript version (line 198ff). We have added the RMSE of the georeferencing as provided in the processing report of the pix4D software (line 209f): The root mean square error (RMSE) of the georeferencing varied between 1.9 cm and 4.7 cm according to the Pix4d processing reports.

*An area of 3x3 pixels seem very small considering geolocation errors. Assuming the plot should be somehow representative of a wider area, would not be more prudent to have a larger window?*

We selected the 3x3 pixel window (approximately 0.3 m x 0.3 m) to most precisely overlap with each subplot (0.25 m x 0.25 m). Due to the high accuracy (0.003 m) of the GNSS measurements and georeferencing (max. 0.047 m) in relation to the window size, we expect just minor location errors. We added this explanation in line 222ff.

Another aspect is that although we aimed to sample homogenous plots, we realized in the field (and by further data analysis) that there were small-scale variations due to different plant species. These small-scale variations would be neglected by a larger window size.

***Results***

*It is somewhat surprising the NIR does not follow DM or height, as this should be rather straightforward (unless for very small range). Is there any factor related to the acquisition that might be causing this issue?*

In Figure 4 of the manuscript we just plotted the spectral profiles of a few selected samples (corresponding to min and max values of DM and N as well as the ones that have DM or N concentration values that approximately correspond to the 25th, 50th and 75th percentile). In Figure 4, the NIR reflectance does follow DM, except for the maximum value. We now created additional figures (Figure 1 below) to show the relationship between the reflectance of the NIR band and DM. In fact, there is a positive relationship between NIR and DM (Spearman's r = 0.49 – 0.55; Figure 1 below). Therefore, the appearance of Figure 4 depends very much on the selection of the percentile.

Potential reasons for the rather weak relationship could be the heterogeneous species compositions of the subplots (the Spearman's correlation between DM and canopy height is r = 0.69; see Figure 3a), differences during the acquisition (time during the day, clouds), as well as the radiometric correction of the multispectral sensors, which can be challenging and a source of uncertainty (Olsson et al., 2021). We included the new plots (Figure 5 in the revised manuscript) and addressed the issue in the discussion of the revised manuscript version (line 591 – 596).

[Figure]

*Figure 1. Scatterplots of NIR reflectance vs. DM for the REM sensor (a) and SEQ sensor (b) with linear model fit (Spearman correlation coefficient and p-value indicated in the plot). Note: there are less data points for REM as there were no flights with this sensor at the Eschenlohe site.*

*The doubt of a strong influence of acquisition factors is also supported by the very poor performance of regression as compared to machine learning and the improved performances on DM including VI in the validation. Also the important role of ground canopy height may suggest that as this variable is clearly not affected by the UAS data acquisition. I would suggest to run some tests and eventually add some considerations in the discussions.*

Thanks for raising these issues. As we conceive the issue, in our data the traits are not strongly correlated to spectral signals and VI in a bivariate comparison (as shown in the new Figure 1 above). The poor performance of the linear models is also largely due to the cross-validation scheme, implying that the pairwise-relationships between a response and a predictor were species and phenological stage dependent. Therefore ML models perform better as they consider multiple predictors at once. Related to that, it has been shown in many studies, that ML approaches outperform linear regression models, so from our perspective this is neither a surprise nor a clear sign of imperfect data acquisition. Also the improvement through the use of VIs is well-confirmed process as VIs are designed to highlight plant physical traits better. In conclusion, the weaker performance of the LM model compared to the ML approaches does not only seem to be because of the acquisition error, but other confounding factors involved as well.

*The word 'notably' is very often repeated. Sometime is a bit redundant*

We replaced or removed part of the "notably" in the text.

*Figure 7. It is a bit strange to see many points along a line (i.e. same N content) in Fig.7b. Is it correct?*

We have checked the figure and it is correctly reflecting the model outcome. The points along a line means that in this model set-up, GBM does not differentiate most of the samples in predicting N concentration. This is because the domain of the input data values of the test site are new to the algorithm, makes the algorithm to guess it is constant. We added an explanation in line 536f.

Yours sincerely,

Anne Schucknecht, on behalf of all co-authors

**Revision notes for referee #2 comment on "Estimating dry biomass and plant nitrogen concentration in pre-Alpine grasslands with low-cost UAS-borne multispectral data – a comparison of sensors, algorithms, and predictor sets" by Anne Schucknecht et al., Biogeosciences Discuss., https://doi.org/10.5194/bg-2021-250-RC2, 2021**

Dear reviewer #2,

Thank you very much for your helpful comments.

You will find our revised manuscript (with track changes) uploaded on the manuscript handling system and below our detailed response to your comments. The original comments from reviewer #2 are in *italic and black* and the responses are in blue. The lines mentioned in the revision note refer to the revised version in track-changes mode of our manuscript.

**Response to reviewer #2**

**Main issue 1 (regarding motivation for comparing two UAS sensors)**

*The motivation for comparing these two UAS sensors is not clear. Are these two types of sensors are popularly used in UAS remote sensing studies? How the findings from the two sensor comparison are relevant to other studies and the UAS remote sensing community?*

We agree with the referee that the motivation of comparing two multispectral sensors is not well addressed in the current manuscript version. Several studies applying low-cost UAS sensors for vegetation mapping/monitoring used the Parrot Sequoia sensor and some highlighted certain quality issues (e.g. Olsson et al, 2021; Poncet et al. 2019). We wanted to test if the Micasense RedEdge-M is a good alternative in the low-cost segment and if it has a better performance due to the additional blue band and slightly different central wavelengths/band widths in the other bands. As the use of (low-cost) UAS sensors is increasing, it is important to learn more about their respective advantages and disadvantages. In the revised version of the manuscript we included some sentences about our motivation to compare two multispectral sensors (line 119 – 123).

*Overall, SEQ and REM sensors are very similar. These two sensors have similar pixel resolution, similar wavelengths in green (550/560 nm), red (660/668 nm), and red edge (735/717 nm). Furthermore, the manuscript pointed out that SEQ performed well for predicting plant nitrogen concentration, while REM had a better performance for predicting dry biomass. However, it is not clear why these two sensors had such different performances in the current manuscript. The analysis and explanation for sensor performance on dry biomass and nitrogen predictions need to be strengthened.*

We agree that the interpretation of the identified difference is a challenge. Note that REM has an additional channel in the blue and its band widths are generally smaller compared to SEQ. The goal of this study is to show potential of (low-cost) UAS sensors, therefore we focused more on the overall performance of the sensors, therefore a detailed comparison of the sensors may be slightly beyond the scope. We strengthened the discussion about the two sensors in chapter 4.3 (line 658 - 666, 679 – 687)

*In Table 2, you labeled 790nm as near infrared. However, we usually refer to 700-800nm as red edge, while wavelengths beyond 800nm as near infrared.*

We use a slightly different terminology closely related to the main principles of vegetation reflectance. As you can see from the absorption spectra shown below, Chlorophyll (a+b) (as well as the Carotenoids) are completely transparent beyond 730nm. Hence, most of the community agrees that the wavelength range between 680 nm to 730 nm is called red-edge – with pigment absorption strongly decreasing with increasing wavelength. According to this definition, beyond 730 nm starts the NIR, where chlorophyll has no longer any effect.

[Figure]

*From the soil-vegetation radiative transfer modeling view, red edge wavelengths are vital for vegetation chlorophyll content and nitrogen content retrieval.*

In principle we agree with your interpretation. But for clarity, please let us not forget that the "optimum wavelengths" for retrieval of pigments depend on their concentration. In other words, if you have high chlorophyll concentrations, you try to move away from the wavelengths with strongest absorption in order to avoid the signal becoming saturated (hence you make more use of red edge wavelengths). However, if your chlorophyll concentrations are (very) low, you try to use bands maximally sensitive to chlorophyll.

*The near infrared is more sensitive to the vegetation canopy structure such as leaf area index and total biomass.*

From physical principles, all spectral bands in the 400-2500nm spectral range are sensitive to LAI (and biomass, which is just LAI x SLA) because more plant material will simply upscale the chlorophyll/nitrogen leave content in g/m2). The retrieval is governed by two facts: (i) spectral difference between background reflectance and leaf reflectance, and (ii) the impact of possible confounding factors. In other words, the advantage of NIR wavelengths (compared for example to visible wavelengths) stems from the fact that in the NIR (i) contrast between soil and leaf reflectance is very high, and (ii) only LAI has an impact (besides average leave angle (ALA) and soil brightness) – but not the highly variable leaf pigments.

*From my interpretation, SEQ has two red edge bands and could potentially get better results for nitrogen concentration retrieval, but not dry biomass as lacking information in near infrared. Meanwhile, REM has information on near infrared which is good for biomass retrieval.*

As pointed out above, we do not see that SEQ has "two red edge bands" if you share our interpretation that those bands are slightly affected by leaf pigments, but to a lesser extent than the NIR. In our opinion, both sensors cover the red edge and the NIR - the main difference being that the REM lacks the blue band.

**Main issue 2 (regarding motivation for selecting GBM and RF)**

*The motivation for selecting Gradient Boosting Machines and Random Forest is also not clear. Why not other more popular machine learning or statistical approaches, such as partial least-squares regression, LASSO, Ridge, or Neural Networks?*

We understand your concern, thanks. Actually, we explored many other algorithms, however the focus of the paper is more on application side, thus comparing a large number of the ML algorithms is beyond that. We constrained ourselves to the two selected algorithms, following a large body of literature which approve those algorithms are often outstanding in performance and robust (citations in Section 2.3.1, line 251f, e.g. Caruana and Niculescu-Mizil, 2006; Fernández-Delgado et al., 2019, 2014; Orzechowski et al., 2018). Furthermore, to our knowledge RF and GBM are as popular as the suggested algorithms including classic Artificial Neural Networks. We did not try Deep Neural Networks, or Deep Learning, which could possibly perform better in this case primarily because of the relatively small number of data points. Instead, we discuss that issue, especially on model transferability in Section (4.6, line 748ff).

*The purpose of applying machine learning algorithms is not only to achieve good model predictive performance. Many machine learning algorithms like random forest can help to identify the relative importance of each feature input. This feature importance analysis is very necessary to understand the relationship between feature inputs and the predicted variables. However, such analysis is missing in this study. I strongly recommend further feature importance analysis to identify scientific linkage among input variables and the predicted variable to strengthen the manuscript result interpretation.*

We agree with the reviewer that a strength of machine learning algorithms is the identification of the relative importance of predictor variables (input features). Actually, we calculated the relative variable importance for both ML for all tested models and provided the detailed results in the supplementary material (Table ST4). Main outcomes were addressed in chapter 3.4.2. In the revised manuscript, we combine the information about variable importance under a subheading of chapter 3.4.2 (line 498 – 513) so that it can be better noticed by the reader.

**Main issue 3 (regarding single date acquisition of UAS and field data)**

*The UAS multispectral data were collected from one single flight in each site. How robustness of these results across different growth stages and dates is uncertain?*

We are aware of the limitation of a single field campaign. The study presented in the manuscript tests the general approach of grassland trait estimation for the application of immediate grassland trait mapping. To increase the general validity or the results of a single campaign, we selected several sites differing in management, species composition, current canopy height and phenological stage, which allowed us to compile a dataset of variable grassland traits. We included this aspect in the material and method section (line 145, 156f, Table 1) and added a paragraph in the discussion chapter (line 733 - 742) to address the applicability of the study and requirements for other phenological stages (multiple campaign applications).

**Main issue 4 (hyper-parameter tuning)**

*Machine learning parameter tuning is a very necessary and common step to implement model training. However, this manuscript highlights the hyper-parameter tuning as one major research question. The innovations of this study need to be strengthened.*

We agree that the importance of ML parameter tuning has been pronounced in many of previous studies, but in practice, we observe many ML applications in geo- and environmental-science applications still omitting a comprehensive calibration procedure. In addition, we tuned using an adaptive algorithm which enabled us to find better parameter values than using a naive grid-searching or latin-hypercube approach. It reduced the time required to calibrate greatly, too. In this paper, we show how an adaptive algorithm can be used in geo- and environmental applications and how much progress actually can be achieved in a practical application, for suggesting that this has to be a mandatory step in similar modelling studies. We revised the discussions to highlight the value of the adaptive algorithms (line 635ff).

**Minor issues**

*There are many abbreviations in Figure 2. The caption should add explanations of these abbreviations for readers.*

We added explanations of the abbreviations in the caption of Figure 2.

*The reflectance values in Figure 4 look quite different from the two sensors. Do you have ground reflectance collection to validate your reflectance?*

Unfortunately, we do not have ground-measured reflectance values for validation. It is important to note that Figure 4 shows the spectral profiles of a subset of the samples for better readability of the plot. An example for all samples is given in the Figure below for NIR reflectance vs. DM (see a comment from referee #1). The figure shows a positive relationship (Spearman correlation coefficient is approx. 0.5) between NIR reflectance and DM, as well as the different reflectance value range of the two sensors. Possible reasons for the different pattern between the REM and SEQ sensor could be the different spectral and radiometric properties,

radiometric calibrations and changes in acquisition conditions. We added further sentences on this in the discussion section of chapter 4.3 (line 679ff).

[Figure]

*Figure 2. Scatterplots of NIR reflectance vs. DM for the REM sensor (a) and SEQ sensor (b) with linear model fit (Spearman correlation coefficient and p-value indicated in the plot). Note: there are less data points for REM as there were no flights with this sensor at the Eschenlohe site.*

*The manuscript mentioned that mountain regions have frequent cloud occurrences to argue the weakness of Copernicus satellite missions. However, UAS data collection under cloudy environment also has data quality issues. The manuscript may need to discuss such potential issues and mitigation strategies.*

We agree with the referee that studies utilizing UAS data have their own challenges and drawbacks (although UAS acquisitions have a higher temporal flexibility). Therefore, we included a section (chapter 4.7) in the discussion chapter that discusses the challenges with UAS and low-cost sensors in general and the limitations of our study in particular.

*Most parts of the manuscript used nitrogen concentration. However, Figure 6 used nitrogen content in the (c) and (d) subplots.*

We corrected Figure 6 (Figure 7 in the revised manuscript).

*The same issue of nitrogen concentration on Figure 7.*

We corrected Figure 7 (Figure 8 in the revised manuscript).

*Figure 8 (d) has clear shadows. The reflectance from these shadows needs to be either corrected to real surface reflectance to quantify vegetation traits or simply removed. I don't think the current estimates for areas in tree shadows are right.*

The estimation of DM and N concentration can just be used for vegetated grassland, not affected by shadows. To make this clearer for the reader, we mentioned in chapter 2.3.8 (line 361f) and in the caption of Figure 8 (Figure 9 in the revised manuscript version) that the plant trait estimates are only valid for un-shaded pixels and vegetated grassland areas.

*The figure panel design of Figure 8 is strange. We normally put RGB into the first subplot. You have paired maps for DM and N. These paired subplots could be in one row.*

We rearrange the sub-figures as proposed.

Yours sincerely,

Anne Schucknecht, on behalf of all co-authors

---

## Author Response (AR2)

**Revision notes for referee #2 comment from 25 Mar 2022 on "Estimating dry biomass and plant nitrogen concentration in pre-Alpine grasslands with low-cost UAS-borne multispectral data – a comparison of sensors, algorithms, and predictor sets" by Anne Schucknecht et al.,**

Dear reviewer #2,

Thank you for your comments regarding the reflectance differences between REM and SEQ.

You will find our revised manuscript (with track changes) uploaded on the manuscript handling system and below our response to your comment. The original comment from reviewer #2 is in *italic and black* and the response is in blue. The lines mentioned in the revision note refer to the revised version of our manuscript.

Response to reviewer #2

*"The authors have thoroughly revised the manuscript and the quality has been significantly improved. However, I am still confused about the significant differences in NIR reflectance between REM and SEQ in Fig. 4. The authors also provided one additional Fig.1 in the response letter. It strikes me that Fig. 1 with all NIR reflectance measurements shows significantly large differences between these two sensors. The NIR reflectance from SEQ can even reach 0.85, which is almost twice of NIR reflectance from REM. The NIR reflectance reaching 0.85 is almost for very dense vegetation conditions not like the case here for grassland. For me, NIR reflectance from REM is much more reasonable. The authors explain that such difference is possibly due to sensor radiometric properties, radiometric calibrations, and changes in acquisition conditions. However, if there are significant uncertainties and errors in these radiometric calibration and data acquisitions, how can we guarantee the quality of reflectance data and their downstream analysis? I suggest to double check your reflectance data. Overall, I suggest a minor revision before accepting this manuscript for publication."*

We agree with the referee that the difference in NIR reflectance between the two sensors is striking and that such high NIR reflectances for the SEQ are indeed not plausible for (grassland) vegetation. We checked the reflectance data, which are ok from the processing site. Furthermore, we adapted Fig. 5 of the manuscript so that it now indicates the plots of the points.

We assume that these high NIR reflectance values are due to a calibration issue of the SEQ sensor. As shown in Figure 1 below (added to the Supplement as Fig. SF1), the SEQ sensor generally shows higher reflectance values than the REM sensor. However, if no transfer learning is involved - hence if we develop and apply models only and solely using this very same sensor - the calibration matters not so much. Though, this would change if we would try to develop models that are to be applied on data acquired by a different sensor. This is not the case for the current study, but would need to be considered in potential future applications. We added these information in the discussion about this issue (line 655ff., line 668ff., line 785ff.).

Furthermore, we did some small language changes in the whole manuscript to correct errors and sometimes readability.

[Figure]

*Figure 1. Reflectance values of REM vs SEQ sensor for different bands.*

Yours sincerely,

Anne Schucknecht, on behalf of all co-authors

---

## Author Response (AR3)

Dear Andreas,

Thank you very much for your constructive comments that we will shortly address in the following.

After using the search function, we realised that we used "better" indeed quite often – in most of the cases in combination with model performance. We agree that this is not fully correct, but would defend the use in combination with performance due to a better readability. As we described the used model evaluation metrics in chapter 2.3.5, we would assume that the reader knows what we mean with "better performance" without the need to write in every sentence "better performance in terms of $R^2$ and RMSE" or something similar. Anyhow, we replaced some of the occurrences of "better performance" with the full description to remind the reader about the used metrics. For the cases related to data quality of a sensor, we agree that using "higher accuracy/precision/robustness/etc." is more appropriate than "better".

With respect to the other two comments, we changed the text as proposed by you.

Additionally, we changed the order of subfigures in the supplement for figures SF11, SF12, SF13 and SF14 according to Fig. 9 in the main manuscript as once suggested by one of the reviewers.

Finally, we would like to thank you for guiding us through the whole review process.

With kind regards,

Anne Schucknecht an behalf of all co-authors